# Viral evasion of the integrated stress response through antagonism of eIF2-P binding to eIF2B

Michael Schoof [1,2,5], Lan Wang [1,2,5], J. Zachery Cogan[1,2], Rosalie E. Lawrence [1,2], Morgane Boone [1,2], Jennifer Deborah Wuerth [3], Adam Frost [2,4] & Peter Walter [1,2✉]

Viral infection triggers activation of the integrated stress response (ISR). In response to viral double-stranded RNA (dsRNA), RNA-activated protein kinase (PKR) phosphorylates the translation initiation factor eIF2, converting it from a translation initiator into a potent translation inhibitor and this restricts the synthesis of viral proteins. Phosphorylated eIF2 (eIF2-P) inhibits translation by binding to eIF2's dedicated, heterodecameric nucleotide exchange factor eIF2B and conformationally inactivating it. We show that the NSs protein of Sandfly Fever Sicilian virus (SFSV) allows the virus to evade the ISR. Mechanistically, NSs tightly binds to eIF2B ($K_D = 30$ nM), blocks eIF2-P binding, and rescues eIF2B GEF activity. Cryo-EM structures demonstrate that SFSV NSs and eIF2-P directly compete, with the primary NSs contacts to eIF2Bα mediated by five 'aromatic fingers'. NSs binding preserves eIF2B activity by maintaining eIF2B's conformation in its active A-State.

[1] Howard Hughes Medical Institute, University of California at San Francisco, San Francisco, CA, USA. [2] Department of Biochemistry and Biophysics, University of California at San Francisco, San Francisco, CA, USA. [3] Institute of Innate Immunity, Medical Faculty, University of Bonn, Bonn, Germany. [4] Chan Zuckerberg Biohub, San Francisco, CA, USA. [5] These authors contributed equally: Michael Schoof, Lan Wang. ✉email: Peter@walterlab.ucsf.edu

The integrated stress response (ISR) is a conserved eukaryotic stress response network that, upon activation by a diverse set of stressors, profoundly reprograms translation. It is coordinated by at least four stress-responsive kinases: PERK (responsive to protein misfolding in the endoplasmic reticulum), PKR (responsive to viral infection), HRI (responsive to heme deficiency and oxidative and mitochondrial stresses), and GCN2 (responsive to nutrient deprivation)[1–4]. All four known ISR kinases converge on the phosphorylation of a single serine (S51) of the α-subunit of the general translation initiation factor eIF2. Under non-stress conditions, eIF2 forms a ternary complex (TC) with methionyl initiator tRNA (Met-tRNAi) and GTP. This complex performs the critical task of delivering the first amino acid to ribosomes at AUG initiation codons. Upon S51 phosphorylation, eIF2 is converted from a substrate to an inhibitor of its dedicated nucleotide exchange factor (GEF) eIF2B. GEF inhibition results from binding of eIF2-P in a new, inhibitory binding orientation on eIF2B, where it elicits allosteric changes to antagonize eIF2 binding and additionally compromise eIF2B's intrinsic enzymatic activity[5,6].

eIF2B is a twofold symmetric heterodecamer composed of two copies each of α, β, δ, γ, and ε subunits[7–10]. eIF2B can exist in a range of stable subcomplexes (eIF2Bβδγε tetramers and eIF2Bα2 dimers) if the concentrations of its constituent subunits are altered[5,8,9,11]. While earlier models suggested eIF2B assembly to be rate-limiting and a potential regulatory step, recent work by us and others show that eIF2B in cells primarily exists in its fully assembled decameric, enzymatically active state[5,6]. Cryo-EM studies of various eIF2B complexes elucidated the mechanisms of nucleotide exchange and ISR inhibition through eIF2-P binding[5,6,12–15]. Under non-stress conditions, eIF2 engages eIF2B through multiple interfaces along a path spanning the heterodecamer. In this arrangement, eIF2α binding to eIF2B critically positions the GTPase domain in eIF2's γ-subunit, allowing for efficient catalysis of nucleotide exchange[12,14]. eIF2B's catalytically active conformation ("A-State") becomes switched to an inactive conformation upon eIF2-P binding (Inhibited or "I-State"), which displays altered substrate-binding interfaces[5,6]. I-State eIF2B(αβδγε)2 exhibits enzymatic activity and substrate engagement akin to the tetrameric eIF2Bβδγε subcomplex; hence, eIF2-P inhibition of eIF2B converts the decamer into conjoined tetramers, which reduces its GEF activity, lowers the cell's TC concentration, and results in ISR-dependent translational reprogramming[5,6].

Viruses hijack the host cell's protein synthesis machinery to produce viral proteins and package new viral particles. Numerous host countermeasures have evolved. In the context of the ISR, double-stranded RNA (dsRNA), a by-product of viral replication, triggers dimerization and autophosphorylation of PKR[3,16]. In this activated state PKR phosphorylates eIF2, which then binds to and inhibits eIF2B. As such, cells downregulate mRNA translation as a strategy to slow the production of virions. Viruses, in turn, enact strategies of evasion. Indeed, viral evasion strategies acting at each step of ISR activation have been observed. Influenza virus, for example, masks its dsRNA[17,18]. Rift Valley Fever virus (RVFV) encodes an effector protein that degrades PKR[19]. Hepatitis C virus blocks PKR dimerization[20]. Vaccinia virus encodes a pseudosubstrate as a PKR decoy[21]. Herpes simplex virus can dephosphorylate eIF2-P[22]. And some coronavirus and picornavirus proteins appear to block the eIF2B–eIF2-P interaction[23]. This evolutionary arms race between host and pathogen can provide invaluable tools and insights into the critical mechanisms of the ISR, as well as other cellular stress responses.

Here, we investigated the previously unknown mechanism by which Sandfly Fever Sicilian virus (SFSV) evades the ISR. SFSV and RVFV are both members of the genus *Phlebovirus* (order Bunyavirales) which encode an evolutionarily related non-structural protein (NSs)[24–26]. Across the phleboviruses, NSs serves to counteract the antiviral interferon response, but NSs proteins perform other functions as well[27,28]. Unlike the RVFV NSs which degrades PKR, SFSV NSs does not impact the levels or phosphorylation status of PKR or eIF2[19,29]. Instead, it binds to eIF2B, inhibiting the ISR. The mechanistic basis of this inhibition was previously unclear. We here provide cellular, biochemical, and structural insight into this question, showing that the SFSV NSs evades all branches of the ISR by binding to eIF2B and selectively blocking eIF2-P binding, thereby maintaining eIF2B in its active A-State.

## Results

**The SFSV NSs is a pan-ISR inhibitor.** To dissect the role of the SFSV NSs (henceforth referred to as NSs) in ISR modulation, we engineered cells stably expressing either an empty vector, a functional NSs (NSs::FLAG), or a non-functional NSs (FLAG::NSs) (Supplementary Fig. 1). As previously reported, the NSs with a C-terminal FLAG tag (NSs::FLAG) should retain its PKR-evading properties while tagging at the N-terminus (FLAG::NSs) blocks this functionality[29]. These constructs were genomically integrated into our previously generated ISR reporter system, in which both changes in ATF4 translation and general translation can be monitored[5]. Both NSs::FLAG and FLAG::NSs were stably expressed in these cells without impacting the levels of key ISR components (eIF2B, eIF2, PKR, PERK) (Fig. 1a). The apparent differences in band intensity between NSs::FLAG and FLAG::NSs may reflect differences in protein stability or, perhaps more likely, differences in antibody affinity for the FLAG epitope at the respective C- and N-terminal tagging locations.

To ask whether NSs is a pan-ISR inhibitor capable of dampening ISR activation irrespective of any particular ISR activating kinase, we chemically activated PERK, HRI, and GCN2 with thapsigargin, oligomycin, and glutamine deprivation/synthetase inhibition through L-methionine sulfoximine, respectively. NSs::FLAG expression dampened the increases in ATF4 translation brought about by activation of any of the kinases (Fig. 1b–d). NSs::FLAG also maintained general translation levels in the thapsigargin and oligomycin treated cells (Fig. 1b, c). Notably, in the context of GCN2 activation, general translation comparably decreased at the highest levels of stress regardless of NSs status (Fig. 1d). This observation likely reflects the additional stress responses that react to reduced amino acid levels, as well as the fact that while the ISR controls translation initiation, ribosome-engaged mRNAs still need sufficient levels of amino acids to be successfully translated. On the whole, these data therefore show that the NSs is a pan-ISR inhibitor akin to the small-molecule ISRIB, which binds to eIF2B and counteracts the ISR by allosterically blocking eIF2-P binding and promoting the eIF2B complex assembly when eIF2B's decameric state is compromised[5,6,30].

**NSs binds decameric eIF2B exclusively.** To explain the mechanism by which NSs inhibits the ISR, we purified NSs expressed in mammalian cells (Fig. 2a, b). We next validated that NSs binds to eIF2B in vitro by immobilizing distinct eIF2B complexes on agarose beads and incubating them with an excess of NSs (Fig. 2c). As expected, NSs binds to the fully assembled eIF2B(αβδγε)2 decamers (Lane 4). Notably, it did not bind to eIF2Bβδγε tetramers (Lane 5) or to eIF2Bα2 dimers (Lane 6). The NSs interaction with eIF2B thus either spans multiple interfaces that are only completed in the fully assembled complex or

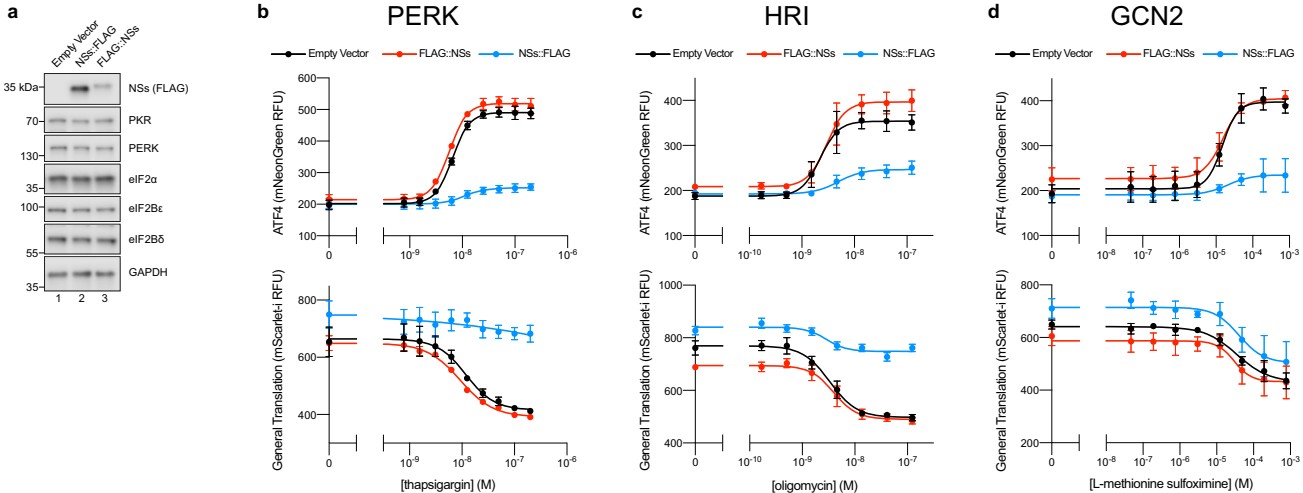

**Fig. 1 The SFSV NSs is a pan-ISR inhibitor. a** Western blot of K562 cell extracts. Loading of all lanes was normalized to total protein. **b–d** ATF4 and General Translation reporter levels as monitored by flow cytometry. Trimethoprim, which is necessary to stabilize the ecDHFR::mScarlet-i and ecDHFR::mNeonGreen translation reporters, was at 20 μM for all conditions. **b** Samples after 3 h of thapsigargin and trimethoprim treatment. **c** Samples after 3 h of oligomycin and trimethoprim treatment. **d** Samples after 4 h of glutamine deprivation, L-methionine sulfoximine, and trimethoprim treatment. For **a**, PERK and GAPDH, PKR and eIF2α, and eIF2Bε and NSs (FLAG) are from the same gels, respectively. eIF2Bδ is from its own gel. For **b–d**, biological replicates: $n = 3$. All error bars represent s.e.m. Source data are provided as a Source Data file.

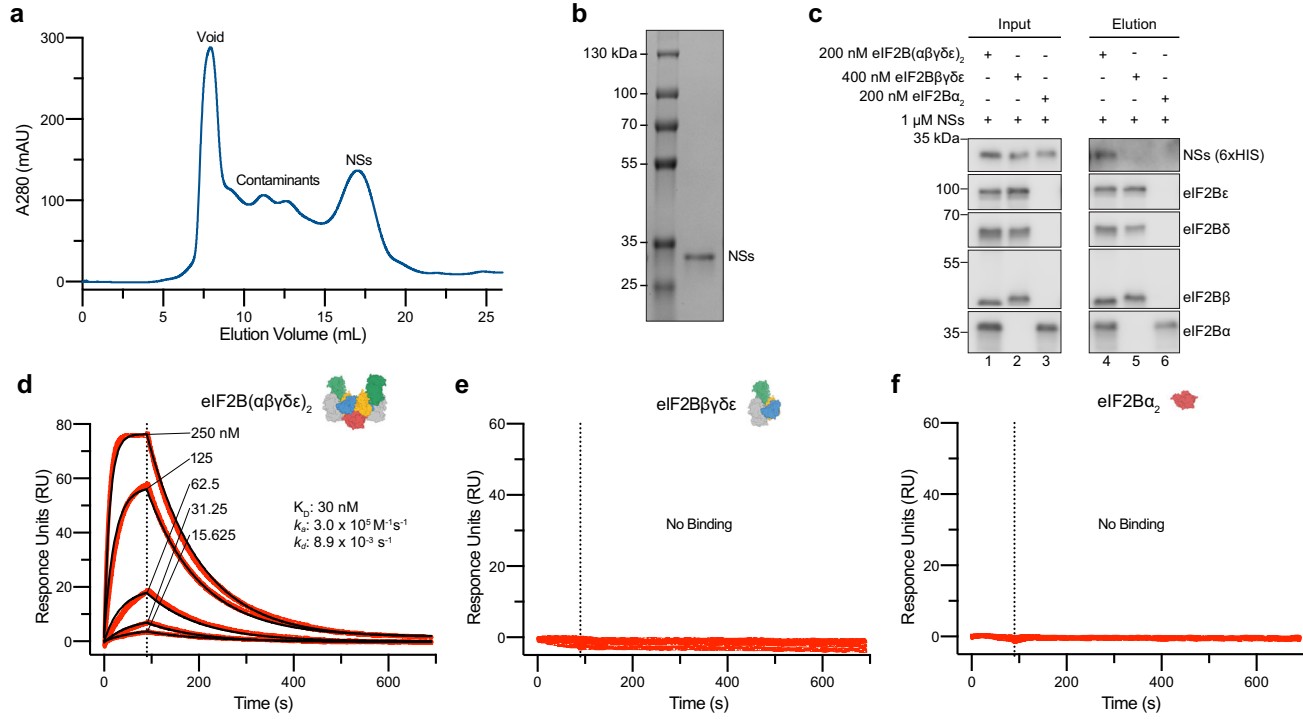

**Fig. 2 NSs specifically binds to eIF2B(αβδγε)₂ decamers. a** Size exclusion chromatogram (Superdex 200 Increase 10/300 GL) during NSs purification from Expi293 cells. **b** Coomassie Blue staining of purified NSs. **c** Western blot of purified protein recovered after incubation with eIF2B(αβγδε)₂, eIF2Bβγδε, or eIF2Bα₂ immobilized on Anti-protein C antibody-conjugated resin. For eIF2B(αβγδε)₂ and eIF2Bα₂, eIF2Bα was protein C tagged. eIF2Bβ was protein C tagged for eIF2Bβγδε. **d–f** SPR of immobilized **d** eIF2B(αβγδε)₂, **e** eIF2Bββδγε, and **f** eIF2Bα₂ binding to NSs. For eIF2B(αβγδε)₂ and eIF2Bββδγε, eIF2Bβ was Avi-tagged and biotinylated. For eIF2Bα₂, eIF2Bα was Avi-tagged and biotinylated. For **d**, concentration series: (250–15.625 nM) For **e**, **f**, concentration series: (125–15.625 nM). For **c**, eIF2Bβ and eIF2Bα, and eIF2Bδ and NSs (6xHIS) are from the same gels, respectively. eIF2Bε is from its own gel. For **b–f**, a single biological replicate. Source data are provided as a Source Data file.

interacts with a region of eIF2B that undergoes a conformational change when in the fully assembled state.

To quantitatively assess NSs binding to eIF2B, we employed surface plasmon resonance (SPR) experiments to determine the affinity of NSs for the various eIF2B complexes (Fig. 2d–f). The

NSs interaction with decameric eIF2B could be modeled using one-phase association and dissociation kinetics. NSs binds to decameric eIF2B with a $K_D$ of 30 nM ($k_a = 3.0 \times 10^5 \, M^{-1} \, s^{-1}$, $k_d = 8.9 \times 10^{-3} \, s^{-1}$) (Fig. 2d). This affinity is comparable to the low nanomolar affinity of ISRIB for decameric eIF2B

(Supplementary Fig. 3)[10]. In this orthogonal approach, we again observed no detectable binding of NSs to eIF2Bβδγε tetramers or eIF2Bα$_2$ dimers (Fig. 2e, f).

**NSs rescues eIF2B activity by blocking eIF2-P binding.** We next sought to explain the mechanism of NSs inhibition of the ISR using our established in vitro systems for studying eIF2B. As is the case with the small-molecule ISRIB, NSs did not impact the intrinsic nucleotide exchange activity of eIF2B as monitored by a fluorescent BODIPY-FL-GDP loading assay (Supplementary Fig. 2). To mimic the conditions during ISR activation, we repeated our nucleotide exchange assay in the presence of the inhibitory eIF2α-P (Fig. 3a). As expected, eIF2α-P inhibited eIF2B GEF activity ($t_{1/2} = 13.4$ min, s.e.m. = 1.5 min), but increasing concentrations of NSs (25 nM: $t_{1/2} = 9.2$ min, s.e.m. = 1.2 min; 100 nM: $t_{1/2} = 6.2$ min, s.e.m. = 0.5 min) overcame the inhibitory effects of eIF2α-P and fully rescued eIF2B GEF activity (uninhibited $t_{1/2} = 6.3$ min, s.e.m. = 0.6 min).

As NSs' ability to affect eIF2B activity markedly manifests in the presence of eIF2α-P, we wondered whether NSs blocks eIF2α-P binding to eIF2B. To test this notion, we utilized a fluorescent ISRIB analog (FAM-ISRIB) that emits light with a higher degree of polarization when bound to eIF2B, compared to being free in solution (Fig. 3b, black and red dots on the $Y$-axis, respectively). It has been previously shown that eIF2α-P binding to eIF2B antagonizes FAM-ISRIB binding by shifting eIF2B into a conformation incapable of binding ISRIB or its analogs (Fig. 3b, blue dot on the $Y$-axis)[5,6]. A titration of NSs into this reaction recovered FAM-ISRIB polarization (EC$_{50} = 72$ nM, s.e.m. = 9 nM), indicating that NSs engages eIF2B and disrupts eIF2α-P's inhibitory binding. To directly show this antagonism, we immobilized eIF2B decamers on agarose beads and incubated with combinations of

NSs and eIF2α-P (Fig. 3c). While individually, both eIF2α-P and NSs bound to eIF2B (Fig. 3c, lanes 4 and 5, respectively), in the presence of saturating NSs, eIF2α-P no longer bound eIF2B (Fig. 3c, lane 6). We next sought to analyze the impact of NSs binding on full-length substrate (eIF2) and inhibitor (eIF2-P) binding through SPR experiments. In this assay we first flowed one analyte over immobilized eIF2B (to saturate the binding site) immediately followed by a mixture of both analytes (to assess whether the second analyte could co-bind elsewhere). Consistent with the nucleotide exchange assay in Fig. 3a, eIF2 and NSs co-bound eIF2B (Fig. 3d, f, increases in RU at 60 s). However, as with the phosphorylated eIF2α subunit alone, the full phosphorylated heterotrimer (eIF2-P) and NSs did not co-bind (Fig. 3e, g, no increases in RU at 60 s). Together, these results demonstrate that the NSs is a potent inhibitor of eIF2-P binding while preserving eIF2 binding.

**NSs binds to eIF2B at the eIF2α-P-binding site and keeps eIF2B in the active A-State.** Having established that the NSs blocks eIF2-P binding to eIF2B, we next assessed whether NSs is an allosteric regulator of eIF2-P binding (as is the case with ISRIB) or, alternatively, whether it directly competes with eIF2-P binding. To answer this question and to rigorously determine NSs' interactions with eIF2B, we turned to cryo-EM. To obtain a homogeneous sample suitable for structural studies, we mixed full-length NSs with decameric eIF2B at a 3:1 molar ratio. We then prepared the sample for cryo-EM imaging and determined the structure of the eIF2B–NSs complex.

Three-dimensional classification with no symmetry assumptions yielded a distinct class of 137,093 particles. Refinement of this class resulted in a map with an average resolution of 2.6 Å (Supplementary Fig. 4). After docking the individual eIF2B subunits into the recorded density, we observed significant extra

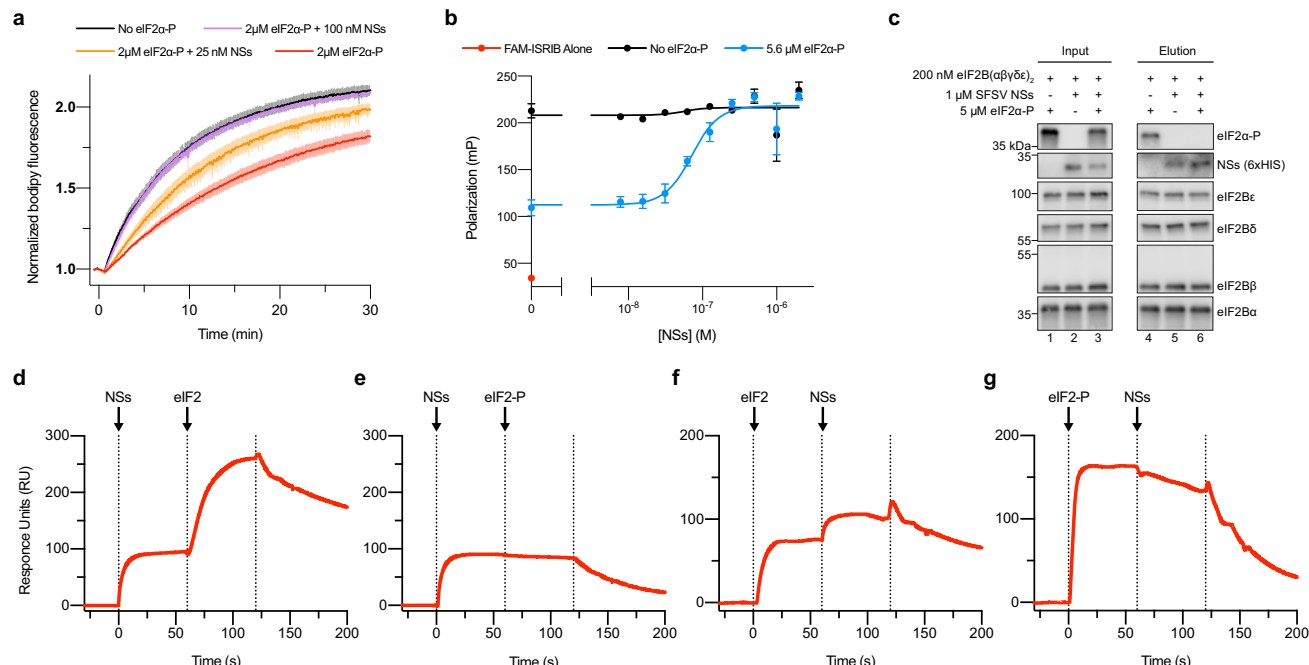

**Fig. 3 NSs grants ISR evasion by antagonizing eIF2α-P binding to eIF2B. a** GEF activity of eIF2B as assessed by BODIPY-FL-GDP exchange. eIF2B(αβδγε)$_2$ at 10 nM throughout. $t_{1/2} = 6.3$ min (No eIF2α-P), 6.2 min (2 μM eIF2α-P + 100 nM NSs), 9.2 min (2 μM eIF2α-P + 25 nM NSs), and 13.4 min (2 μM eIF2α-P). **b** Plot of fluorescence polarization signal before (red) and after incubation of FAM-ISRIB (2.5 nM) with 100 nM eIF2B(αβδγε)$_2$ (black) or 100 nM eIF2B(αβδγε)$_2$ + 5.6 μM eIF2α-P (blue) and varying concentrations of NSs. **c** Western blot of purified protein recovered after incubation with eIF2B(αβδγε)$_2$ immobilized on Anti-protein C antibody-conjugated resin. eIF2Bα was protein C tagged. **d–g** SPR of immobilized eIF2B(αβδγε)$_2$ binding to saturating **d**, **e** 500 nM NSs, **f** 125 nM eIF2, or **g** 125 nM eIF2-P followed by **d** 125 nM eIF2, **e** 125 nM eIF2-P, or **f**, **g** 500 nM NSs. eIF2Bα was Avi-tagged and biotinylated. For **c**, eIF2Bε and eIF2α-P, eIF2Bβ and eIF2Bα, and eIF2Bδ and NSs (6xHIS) are from the same gels, respectively. For **a**, **b**, biological replicates: $n = 3$. For **c–g**, a single biological replicate. All error bars represent s.e.m. Source data are provided as a Source Data file.

density next to both eIF2Bα subunits, indicating that two copies of NSs are bound to each eIF2B decamer (Fig. 4a and Supplementary Fig. 4). The local resolution of the NSs ranges from 2.5 Å (regions close to eIF2B) to >4.0 Å (periphery), with most of the side chain densities clearly visible (Supplementary Fig. 4). To build the molecular model for NSs, we split the protein into two domains. The C-terminal domain was built using the crystal structure of the C-terminal domain of the RVFV NSs (PDB ID: 5OOO) as a homology model (43.8% sequence similarity with the C-terminal domain of the SFSV NSs (residues 85–261)) (Supplementary Fig. 5)[31]. The N-terminal domain of the NSs (residues 1–84) was built de novo (Supplementary Table 1). The high-resolution map allowed us to build a model for the majority of NSs. The map quality of both NSs molecules are comparable, and their molecular models are nearly identical (root mean square deviation (RMSD) ≈ 0.2 Å). We henceforth focus our analysis on one of them (chain K).

Two copies of NSs bind to one decameric eIF2B in a symmetric manner (Fig. 4a). An overlay of the NSs-bound eIF2B and the eIF2α-P-bound eIF2B structures (PDB ID: 6O9Z) shows a significant clash between the NSs and eIF2-αP, indicating that, unlike the allosteric regulator ISRIB, NSs binds in direct competition with eIF2α-P (Fig. 4d–f). Interestingly, whereas eIF2α-P forms extensive interactions with both the α and the δ subunits of eIF2B, the NSs mainly interacts with the eIF2Bα subunit. The expansive interactions between eIF2α-P and both eIF2Bα and eIF2Bδ mediate a shift in eIF2B's conformation from eIF2B's enzymatically active A-State to its inhibited I-State[5,6]. Thus, despite binding to a region known to influence eIF2B's conformation, an overlay of the NSs-bound eIF2B and apo eIF2B shows that the overall conformation of eIF2B in the two structures are virtually identical (Fig. 4b). By contrast, the eIF2B–NSs and eIF2B–eIF2α-P overlay shows major

conformational differences (Fig. 4c). Together, these structural data, paired with our in vitro assays, show that the NSs grants SFSV evasion of the ISR by directly competing off eIF2-P and restoring eIF2B to its enzymatically active A-State.

## NSs uses a novel protein fold containing aromatic fingers to bind eIF2B

Next, we sought to interrogate the molecular details of the NSs–eIF2B interaction. As mentioned above, NSs consists of two domains. Its N-terminal domain (amino acids 1–84) consists of six β-strands and interacts directly with eIF2B. A search in the DALI protein structure comparison server did not reveal any hits, suggesting a novel protein fold. β-Strands 1 and 2 and β-strands 3 and 4 form two antiparallel β-sheets and fold on top of the C-terminal domain (Supplementary Fig. 6b). The C-terminal domain (amino acids 85–261) is largely α-helical and presumably supports the folding of the N-terminal domain, as truncating the C-terminal domain results in the complete loss of NSs activity in terms of ISR evasion (Supplementary Fig. 7). Also, despite the moderate sequence conservation of the C-terminal domain of the SFSV NSs and the RVFV NSs, their structures overlay extensively (RMSD ≈ 0.2 Å, Supplementary Fig. 6).

The surface of the N-terminal domain forms a hand shape that grips the alpha helices of eIF2Bα, akin to a koala grabbing a eucalyptus branch (Fig. 5a and Supplementary Fig. 10). In this arrangement, the N-terminal domain extends three loops that contact eIF2Bα. The first two loops sit in a groove between helices α3 and α4 and the third loop just below helix α3, effectively sandwiching helix α3 (Fig. 5b). Together, the three loops extend five aromatic amino acids to contact eIF2Bα. We refer to these aromatic amino acids as "aromatic fingers". On the top side of helix α3, the side chain of NSs Y5 forms a cation–π interaction with eIF2Bα R74 and its backbone carbonyl forms a hydrogen

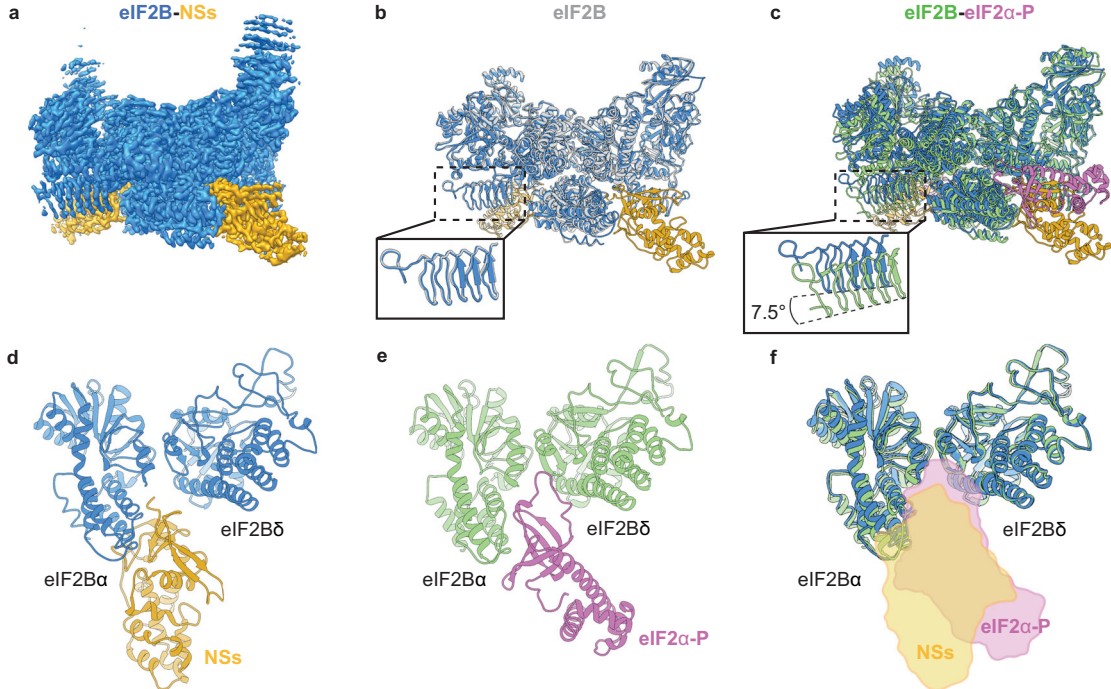

**Fig. 4 Overall architecture of the eIF2B–NSs complex. a** Cryo-EM map of the eIF2B–NSs complex. **b** Overlay of the apo eIF2B structure (PDB ID: 7L70) and the eIF2B–NSs structure shows that the overall conformation of eIF2B is nearly identical between the NSs-bound state and the apo state. **c** Overlay of the eIF2B–eIF2α-P complex structure (PDB ID: 6O9Z) and the eIF2B–NSs structure shows a 7.5° hinge movement between the two eIF2B halves. **d**, **e** Both NSs and eIF2α-P bind to eIF2B at the cleft between eIF2Bα and eIF2Bδ. **d** NSs mainly contacts eIF2Bα, whereas **e** eIF2α-P makes extensive contacts to both eIF2Bα and eIF2Bδ. **f** Comparison between the surfaces of NSs and eIF2α-P showing a significant overlay between the two. eIF2B in the eIF2B–NSs complex is colored in blue and NSs in gold. eIF2B in its apo form is colored white. eIF2B in the eIF2α-P-bound complex is colored in green, and eIF2α-P in pink.

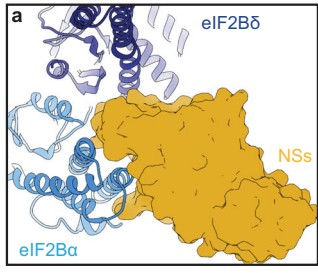

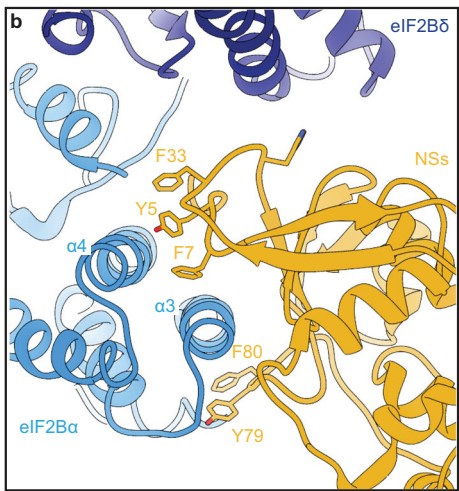

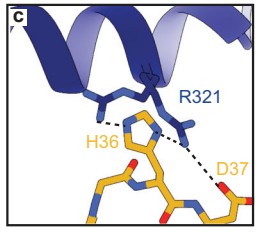

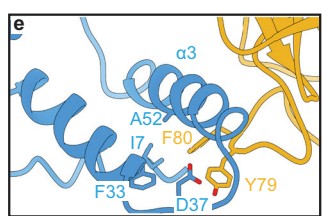

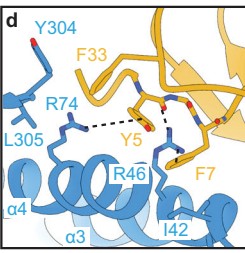

**Fig. 5 NSs latches on to eIF2B with its aromatic fingers. a** Surface representation of NSs showing that it grips the alpha helices of eIF2Bα. **b** NSs extends five aromatic amino acids in three short loops to contact eIF2Bα. They contact helices α3 and α4 of eIF2Bα. The backbone of T35 and the side chains of H36 and D37 of NSs make contact with eIF2Bδ. **c** Zoomed-in view of panel **b** showing the interaction between H36 and D37 with eIF2Bδ. **d**, **e** Zoomed-in view of panel **b** showing the detailed interactions between the five main aromatic amino acids and eIF2Bα. Each polar–polar or cation–π interaction is denoted by a dashed line. NSs is colored in gold, eIF2Bα in blue, and eIF2Bδ in purple.

bond with eIF2Bα R46 (Fig. 5d). NSs F7 forms a cation–π interaction with eIF2Bα R46, and hydrophobic stacking with eIF2Bα I42. NSs F33 stacks against the backbone of eIF2Bα Y304 and L305, as well as the aliphatic region of eIF2Bα R74. On the bottom side of helix α3, NSs F80 stacks against a hydrophobic groove formed by eIF2Bα I7, F33, and A52 (Fig. 5e). NSs Y79 forms a polar interaction with eIF2Bα D37, completing the extensive interaction network of the NSs' aromatic fingers with the α helices in eIF2Bα. In addition, the side chain of NSs H36 and the backbone carbonyl of NSs T35 both contact eIF2Bδ R321. The side chain of NSs D37 also forms an ionic interaction with eIF2Bδ R321, although the distance is close to 4.0 Å, suggesting a weak interaction. These three amino acids account for the only interactions with eIF2Bδ (Fig. 5c).

To validate the functional importance of the eIF2Bα-facing aromatic fingers, we mutated them in pairs or singly to alanines (Y5A/F7A, Y79A/F80A, and F33A) and stably expressed these NSs variants in the dual ISR reporter cells. The point mutations did not compromise NSs stability and, as with WT NSs, did not affect eIF2 or eIF2B subunit levels (Fig. 6a). Upon stress, eIF2α became phosphorylated in all cell lines, but only in cells expressing WT NSs::FLAG was ATF4 translation blunted (Fig. 6a). A similar picture emerged from analysis of the fluorescent ISR reporter signals. Whereas WT NSs inhibited the translation of ATF4 and maintained general translation at roughly normal levels, all the point mutants tested broke the NSs' function as an ISR evader (Fig. 6b). All five eIF2Bα-facing aromatic fingers thus appear critical for NSs modulation of the ISR, likely through reducing the binding affinity of NSs for eIF2B. Indeed, alanine substitutions of the aromatic fingers was independently shown to reduce NSs binding affinity to eIF2B[32].

We additionally assessed the importance of the eIF2Bδ-facing residues—generating stable lines with alanine mutations (H36A and D37A). As we saw with mutation of the aromatic fingers, neither H36A nor D37A impaired NSs translation or impacted eIF2 or eIF2B subunit levels, but ISR evasion as monitored by ATF4 translation became compromised (Fig. 6c). Notably, NSs::FLAG (H36A) displayed an intermediate phenotype in the ATF4 and general translation reporter assays, suggesting that while this mutation compromises NSs binding it does not appear to entirely break the interaction (Fig. 6d). In contrast, NSs::FLAG

(D37A)-expressing cells appear unable to resist ISR activation. Although the structure suggests only a mild ionic interaction between NSs D37 and eIF2Bδ R321, we reason the D37A mutation might not only break the ionic interaction, but also potentially alter the conformation of the loop. As a result, V38 would move, disrupting its stacking with M6, an amino acid next to two aromatic fingers (Y5 and F7) (Supplementary Fig. 8). Thus, changes to D37 and H36 could result in the repositioning of the eIF2Bα-facing aromatic fingers, leading to a complete loss of NSs interaction with eIF2B. Together, these data provide a rationale for NSs' potent and selective binding to only fully assembled eIF2B(αβδγε)₂ decamers.

## Discussion

As one of the strategies in the evolutionary arms race between viruses and the host cells they infect, mammalian cells activate the ISR to temporarily shut down translation, thus preventing the synthesis of viral proteins. Viruses, in turn, have evolved ways to evade the ISR, typically by disarming the PKR branch through countermeasures that lead to decreased levels of eIF2-P, thus allowing translation to continue. In this study, we show that SFSV expresses a protein (NSs) that allows it to evade not just PKR-mediated ISR activation, but all four branches of the ISR, through a mechanism that exploits the conformational flexibility of eIF2B. NSs is an antagonist of eIF2B's inhibitor eIF2-P, deploying an overlapping binding site. Whereas eIF2-P shifts eIF2B to its inactive I-State conformation by closing the angle between the eIF2Bα and eIF2Bδ subunits, NSs engages the enzyme to opposite effect, binding to an overlapping site with eIF2-P but preserving the angle between eIF2Bα and eIF2Bδ and locking it into its active A-State conformation (Fig. 7).

Previously, we and others showed that the GEF activity of eIF2B is modulated conformationally: eIF2B's substrate (eIF2) binding stabilizes it in the A-State, whereas its inhibitor (eIF2-P) binding induces a hinge motion between the two tetrameric halves, resulting in a conformation that cannot engage the substrate optimally (I-State)[5,6]. Our structure shows that NSs antagonizes the endogenous inhibitor (eIF2-P) by directly competing it off and stabilizing eIF2B in the active conformation. Owing to the reported single digit nM affinity of eIF2-P for eIF2B,

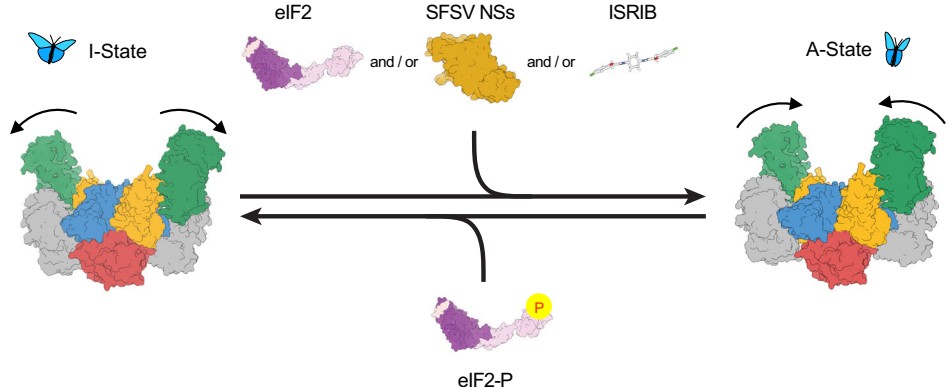

**Fig. 6 All 5 aromatic fingers are required for NSs evasion of the ISR. a, c** Western blot of K562 cell extracts 3 h after treatment with 50 nM thapsigargin. Loading of all lanes was normalized to total protein. **b, d** ATF4 and General Translation reporter levels as monitored by flow cytometry after 3 h of thapsigargin and trimethoprim (20 μM) treatment. For **a**, ATF4 and eIF2α, eIF2Bε and NSs (FLAG), and eIF2Bδ and eIF2α-P are from the same gels, respectively. GAPDH is from its own gel. For **c**, ATF4 and GAPDH, eIF2Bε and NSs (FLAG), and eIF2Bδ and eIF2α-P are from the same gels, respectively. eIF2α is from its own gel. For **b**, biological replicates: $n = 3$. For **d**, biological replicates: $n = 4$. All error bars represent s.e.m. Source data are provided as a Source Data file.

**Fig. 7 Model for regulation of eIF2B activity.** Like the small-molecule ISRIB and the substrate eIF2, NSs binds to and stabilizes the active, "wings up" conformation of eIF2B (A-State). eIF2-P induces the inhibited "wings down" conformation of eIF2B (I-State).

this likely entails a cellular excess of NSs relative to eIF2-P (which should be expected given the high levels at which viral proteins are typically expressed)[33–35]. While NSs binds to the inhibitor-binding site, it does not induce the conformational change that the inhibitor binding induces. This mechanism is reminiscent of the antagonistic inhibition of GPCRs, such as the β-adrenergic receptors, where binding of an agonist ligand shifts the receptor to its active conformation, whereas binding of an antagonist ligand occupies an overlapping but not identical binding site that lacks contacts required to induce the activating conformational change[36–39]. NSs, however, is an antagonist of an inhibitor (eIF2-P). Thus, by inhibiting an inhibition, it actually works as an eIF2B activator under conditions where eIF2-P is present and the ISR is induced.

In its ability to modulate eIF2B, NSs is not unique among viral proteins. The beluga whale coronavirus (Bw-CoV) protein AcP10 likewise allows evasion of the host cell ISR by interacting with eIF2B, as does the picornavirus AiVL protein[23]. It was suggested that AcP10 makes contacts with eIF2Bα and eIF2Bδ, akin to NSs, and hence may act through a similar mechanism by antagonizing eIF2-P, although no structural information is yet available. By primary sequence comparison, AcP10, AiVL, and NSs show no recognizable homology with one another, indicating that viruses have evolved at least three—and likely more—different ways to exploit the eIF2α-P binding site on eIF2B to shut off the ISR. Therefore, inhibiting the eIF2B–eIF2-P interaction through the antagonism of eIF2-P binding could also be a general strategy used by many viruses.

Our structure and mutational analysis suggest that the binding of different parts of NSs to eIF2B occurs in a highly synergistic manner. While the amino acids facing eIF2Bδ do not seem to make sufficiently intimate contacts to provide a significant contribution to the enthalpic binding energy, changing them disrupts binding. It is plausible that the contacts of NSs with eIF2Bδ allow the optimal positioning of the aromatic fingers through allosteric communications between the loops and thus license NSs for tight binding.

The structure of the eIF2B–NSs complex reveals a previously unknown site on eIF2B that is potentially druggable. Unlike ISRIB, which stabilizes eIF2B's A-State through binding to a narrow pocket at the center of eIF2B and stapling the two tetrameric halves together at a precise distance and angle, NSs binds to a different interface on the opposite side of the protein. With ISRIB-derivatives showing extreme promise to alleviate cognitive dysfunction in animal studies of various neurological disorders and recently progressing into the clinic for Phase I human trials, developing therapeutics that modulate the ISR has never been more relevant[40].

Across phleboviruses, all characterized members of the family of related NSs proteins also counteract the host's interferon response[25,26]. For RVFV, this functionality is contained within the structurally conserved C-terminal domain, which nonetheless varies quite heavily in sequence space[27,28,41,42]. A strict functional conservation does not appear to be the case for the N-terminal domain. Although this domain serves to evade PKR in some phleboviruses such as RVFV and SFSV, it accomplishes it through entirely different means: degradation of PKR in RVFV and antagonism of eIF2-P binding to eIF2B in SFSV[19,29]. The NSs is thus a bispecific molecule—a multitool of sorts. The C-terminal domain may serve as a scaffold containing a core functionality upon which the N-terminal domain may be free to evolve, exploring diverse functionalities and mechanisms. It is exciting to speculate whether anti-PKR properties of the N-terminal domain, as we identified for SFSV NSs, are commonly found across phleboviruses and whether still other PKR evasion strategies can be found.

Aberrant ISR activation underlies many neurological disorders (traumatic brain injury, Down's syndrome, Alzheimer's disease, amyotrophic lateral sclerosis), as well as certain cancers (metastatic prostate cancer)[40,43–47]. Virotherapy, where viruses are used as a therapeutic agent for particular diseases, has seen the most success in the realm of cancer treatment where the infection either directly attacks cancer cells (oncolytic virotherapy) or serves to activate host defenses which target virus and cancer alike[48,49]. Indeed, decades of evidence have shown that cancer patients who experience an unrelated viral infection can show signs of improvement, paving the way for the generation of genetically engineered oncolytic viruses that have only just received FDA approval in the last decade[50,51]. With our ever-growing understanding of diverse host–virus interactions, a whole host of new virotherapies are imaginable that can exploit the evolved functionalities of viral proteins such as the NSs.

## Methods

**Cloning of NSs expression plasmids.** The NSs::6xHIS Expi293 expression plasmid for transient transfection was generated using In-Fusion HD cloning. The SFSV NSs sequence[29] was inserted into the pXSN vector backbone and a 6xHIS tag was added at the C-terminus. The various NSs overexpression plasmids for stable lentiviral integration were generated using In-Fusion HD cloning. The SFSV NSs sequence was inserted into the pDBR vector backbone and a FLAG tag was added at the C-terminus (pMS110, pMS127, pMS128, pMS129, pMS130, pMS131, pMS132, pMS133) or N-terminus (pMS111). The various NSs truncations did not have a FLAG tag (pMS119, pMS120, pMS121, pMS122, pMS123). An empty vector control plasmid with no NSs insertion was also generated (pMS085). An IRES followed by the puromycin resistance gene, a T2A self-cleaving peptide, and the BFP sequence allows for selection based on antibiotic resistance or BFP signal (what was used in this study) (Supplementary Fig. 1). Full plasmid details are shown in Supplementary Table 2.

**Cloning of tagged human eIF2B expression plasmids.** *eIF2B2* (encoding eIF2Bβ) and *eIF2B4* (encoding eIF2Bδ) had previously been inserted into sites 1 and 2 of pACYCDuet-1, respectively (pJT073)[8]. In-Fusion HD cloning (Takara Bio) was used to edit this plasmid further and insert an Avi tag (GLNDIFEAQKIEWHE) or a Protein C tag (EDQVDPRLIDGK) at the N-terminus of *eIF2B2*, immediately following the pre-existing 6xHIS tag (pMS001 and pMS003). *eIF2B1* (encoding eIF2Bα) had previously been inserted into site 1 of pETDuet-1 (pJT075)[8]. In-Fusion HD cloning was used to edit this plasmid further and insert an Avi tag at the N-terminus of *eIF2B1*, immediately following the pre-existing 6xHIS tag (pMS026). The Avi tag allows selective, single, and complete biotinylation of the tagged protein.

**Generation of stable NSs-expressing cells in an ISR reporter cell line.** Our previously generated dual ISR reporter K562 cells expressing a stably integrated ATF4 reporter (pMS086), general translation reporter (pMS078), and dCas9-KRAB were used as the parental line[5]. The various NSs overexpression constructs (Supplementary Table 2) were integrated using a lentiviral vector. Vesicular stomatitis virus (VSV)-G pseudotyped lentivirus was prepared using standard protocols and 293METR packaging cells. Viral supernatants were filtered (0.45 μm low protein binding filter unit (EMD Millipore)) and concentrated 10–20-fold (Amicon Ultra-15 concentrator with a 100,000-Da molecular mass cutoff). Concentrated supernatant was then used the same day or frozen for future use. For spinfection, approximately 1,000,000 K562 cells were mixed with concentrated lentivirus and fresh media (RPMI containing 4.5 g/l glucose and 25 mM HEPES supplemented with 10% FBS, 2 mM L-alanyl-L-glutamine (Gibco GlutaMAX), and penicillin/streptomycin), supplemented with polybrene to 8 μg/ml, brought to 1.5 ml in a six-well plate, and centrifuged for 1.5 h at 1000g. Cells were then allowed to recover and expand for ~1 week before sorting on a Sony SH800 cytometer to isolate cells that had integrated the reporter. Roughly 100,000 BFP-positive cells (targeting the highest 1–3% of expressers) were then sorted into a final pooled population and allowed to recover and expand. Cells expressing NSs truncations (pMS119-pMS123) were not sorted and instead analyzed as a polyclonal population, gating for BFP-positive cells during data analysis.

**Western blotting.** Western blotting was performed as previously described[5]. In brief, approximately 1,000,000 cells of the appropriate cell type were drugged as described in individual assays and then pelleted, washed, pelleted again, and resuspended in lysis buffer. Cells were then rotated for 30 min at 4 °C and then spun at 12,000g for 20 min to pellet cell debris. Protein concentration was measured using a bicinchoninic acid assay (BCA assay) and within an experiment, total protein concentration was normalized to the least concentrated sample. Equal protein content for each condition (targeting 10 μg) was run on 10%

Mini-PROTEAN TGX precast protein gels (Biorad). After electrophoresis, samples were transferred onto a nitrocellulose membrane. Primary antibody/blocking conditions for each protein of interest are outlined in Supplementary Table 3. Membranes were developed with SuperSignal West Dura (Thermo Fisher Scientific). Developed membranes were imaged on a LI-COR Odyssey gel imager for 0.5–10 min depending on band intensity.

**ATF4/general translation reporter assays.** ISR reporter cells (at ~500,000/ml) were co-treated with varying combinations of drugs (20 μM trimethoprim plus one of the following: thapsigargin, oligomycin, or glutamine deprivation (and no FBS) + L-methionine sulfoximine) and incubated at 37 °C until the appropriate timepoint had been reached. At this time, the plate was removed from the incubator and samples were incubated on ice for 10 min. Then ATF4 (mNeonGreen) and General Translation (mScarlet-i) reporter levels were monitored using a high throughput sampler (HTS) attached to a BD FACSCelesta cytometer running BD FACSDiva v9.0. Data were analyzed in FlowJo version 10.6.1, and median fluorescence values for both reporters were exported and plotted in GraphPad Prism 8 (Supplementary Fig. 9). No BFP-positive sorting was performed on the lines expressing NSs truncations. For analysis of these samples, BFP-positive cells were gated in FlowJo and analysis performed on this population. Where appropriate, curves were fit to log[inhibitor] vs response function with variable slope.

**Purification of human eIF2B subcomplexes.** Human eIFBα$_2$ (pJT075), Avi-tagged eIFBα$_2$ (pMS026), protein C-tagged eIFBα$_2$ (pMS027), eIF2Bβγδε (pJT073 and pJT074 co-expression), Avi-tagged eIF2Bβγδε (pMS001 and pJT074 co-expression), and Protein C-tagged eIF2Bβγδε (pMS003 and pJT074 co-expression) were purified as previously described with a minor modification for purification of the Avi-tagged species[8]. One Shot BL21 Star (DE3) chemically competent E. coli cells (Invitrogen) were transformed with the requisite expression plasmids and grown in LB with kanamycin and chloramphenicol (eIF2B tetramer preps) or ampicillin (eIF2Bα$_2$ preps). At an OD$_{600}$ of ~0.8 1 mM IPTG (Gold Biotechnology) was added and the culture was grown overnight at 16 °C. Using the EmulsiFlex-C3 (Avestin), Cells were harvested and lysed through three cycles of high-pressure homogenization in lysis buffer (20 mM HEPES-KOH, pH 7.5, 250 mM KCl, 1 mM dithiothreitol (DTT), 5 mM MgCl$_2$, 15 mM imidazole, and cOmplete EDTA-free protease inhibitor cocktail (Roche)). For eIF2Bα$_2$ preps 20 mM imidazole was used. The lysate was clarified at 30,000g for 30 min at 4 °C. Lysate was then clarified at 30,000g for 60 min at 4 °C.

All following purification steps were conducted on the ÄKTA Pure (GE Healthcare) system at 4 °C. Clarified lysate was loaded onto a 5 ml HisTrap HP column (GE Healthcare). For eIF2B tetramer preps the column was then washed in a buffer containing 20 mM HEPES-KOH, pH 7.5, 200 mM KCl, 1 mM DTT, 5 mM MgCl$_2$, and 15 mM imidazole. For eIF2Bα$_2$ preps 30 mM KCl and 20 mM imidazole were used. The sample was then eluted with a linear gradient up to 300 mM imidazole. eIF2B containing fractions were collected and applied to a MonoQ HR 10/100 GL column (GE Healthcare) equilibrated in 20 mM HEPES-KOH pH 7.5, 200 mM KCl, 1 mM DTT, and 5 mM MgCl$_2$. For eIF2Bα$_2$ preps 30 mM KCl was used. The column was washed in the same buffer, and the protein was eluted with a linear gradient up to 500 mM KCl. eIF2B containing fractions were collected and concentrated with an Amicon Ultra-15 concentrator (EMD Millipore) with a 30 kDa (tetramer preps) or 10 kDa (eIF2Bα$_2$ preps) molecular mass cutoff and spun down for 10 min at 10,000g to remove aggregates. The supernatant was then injected onto a Superdex 200 10/300 GL (GE Healthcare) column equilibrated in a buffer containing 20 mM HEPES-KOH pH 7.5, 200 mM KCl, 1 mM DTT, 5 mM MgCl$_2$, and 5% glycerol, and concentrated using the appropriate Amicon Ultra-15 concentrators (EMD Millipore).

For Avi-tagged species, after running samples over a MonoQ HR 10/10 column the eluted fractions were combined and concentrated to a target concentration of 40 μM. The sample was then incubated at 4 °C overnight according to the manufacturer's instructions with 2.5 μg BirA for every 10 nmol substrate, 10 mM ATP, 50 μM d-biotin, and 100 mM Mg(OAc)$^2$ in a 50 mM bicine buffer, pH 8.3 (Avidity BirA biotin-protein ligase standard reaction kit). Incubation with BirA yields selective and efficient biotinylation of Avi-tagged species. After the biotinylation reaction, purification of biotinylated species proceeded as described above.

All eIF2B(αβγδε)$_2$ used throughout was assembled by mixing purified eIF2Bβγδε and eIF2Bα$_2$ (either tagged or untagged versions as needed) at the appropriate molar ratios.

**Purification of human eIF2αβγ heterotrimer and eIF2α-P.** Human eIF2 was purified as previously described[52]. This material was a generous gift of Calico Life Sciences LLC. eIF2-P was prepared by mixing eIF2 in 50-fold excess with PERK kinase and 1 mM ATP. The mixture was incubated at room temperature for 60 min before incubation on ice until use. The purification of human eIF2α-P was performed as previously described[5]. One Shot BL21 Star (DE3) chemically competent E. coli cells (Invitrogen) were transformed with the expression plasmid for N-terminally 6x-His-tagged human eIF2α (pAA007) along with a tetracycline-inducible, chloramphenicol-resistant plasmid (pG-Tf2) containing the chaperones groES, groEL, and Tig (Takara Bio). Transformed cells were grown in LB with

kanamycin and chloramphenicol for selection. Chaperone expression was induced at an OD$_{600}$ of ~0.2, by addition of tetracycline (1 ng/ml). At an OD$_{600}$ of ~0.8 the culture was cooled to room temperature and eIF2α expression was induced with 1 mM IPTG (Gold Biotechnology) and the culture was grown for at least 16 h more at 16 °C. Using the EmulsiFlex-C3 (Avestin), cells were harvested and lysed through three cycles of high-pressure homogenization in lysis buffer (100 mM HEPES-KOH, pH 7.5, 300 mM KCl, 2 mM dithiothreitol (DTT), 5 mM MgCl$_2$, 5 mM imidazole, 10% glycerol, 0.1% IGEPAL CA-630, and cOmplete EDTA-free protease inhibitor cocktail (Roche)). The lysate was clarified at 30,000g for 30 min at 4 °C.

Subsequent purification steps were conducted on the ÄKTA Pure (GE Healthcare) system at 4 °C. Clarified lysate was loaded onto a 5 ml HisTrap FF Crude column (GE Healthcare), washed in a buffer containing 20 mM HEPES-KOH, pH 7.5, 100 mM KCl, 5% glycerol, 1 mM DTT, 5 mM MgCl$_2$, 0.1% IGEPAL CA-630, and 20 mM imidazole, and eluted with 75 ml linear gradient of 20–500 mM imidazole. The eIF2α-containing fractions were collected and applied to a MonoQ HR 10/100 GL column (GE Healthcare) equilibrated in anion exchange buffer (20 mM HEPES-KOH pH 7.5, 100 mM KCl, 1 mM DTT, 5% glycerol, and 5 mM MgCl$_2$). The column was washed in the same buffer, and the protein was eluted with a linear gradient of 100 mM to 1 M KCl. eIF2α containing fractions were collected and concentrated with an Amicon Ultra-15 concentrator (EMD Millipore) with a 30 kDa molecular mass cutoff and spun down for 10 min at 10,000g to remove aggregates. Before size exclusion, the pooled anion exchange fractions were phosphorylated in vitro overnight at 4 °C with 1 mM ATP and 1 μg of PKR$_{(252–551)}$-GST enzyme (Thermo Scientific) per mg of eIF2α. The supernatant was then injected onto a Superdex 75 10/300 GL (GE Healthcare) column equilibrated in a buffer containing 20 mM HEPES-KOH pH 7.5, 100 mM KCl, 1 mM DTT, 5 mM MgCl$_2$, and 5% glycerol, and concentrated using Amicon Ultra-15 concentrators (EMD Millipore) with a 10 kDa molecular mass cutoff. Complete phosphorylation was confirmed by running the samples on a 12.5% Super-Sep PhosTag gel (Wako Chemicals).

**Purification of NSs::6xHIS.** We used the pMS113 construct to express and purify NSs::6xHIS. Expi293T cells (Thermo Fisher) were transfected with the NSs construct per the manufacturer's instructions for the MaxTiter protocol and harvested 5 days after transfection. Cells were pelleted (1000g, 4 min) and resuspended in Lysis Buffer (130 mM KCl, 2.5 mM MgCl$_2$, 25 mM HEPES-KOH pH 7.4, 2 mM EGTA, 1% triton, 1 mM TCEP, 1× cOmplete protease inhibitor cocktail (Roche)). Cells were then incubated for 30 min at 4 °C and then spun at 30,000g for 1 h to pellet cell debris. Lysate was applied to a 5 ml HisTrap HP column (GE Healthcare) equilibrated in Buffer A (20 mM HEPES-KOH, pH 7.5, 200 mM KCl, 5 mM MgCl$_2$, 15 mM imidazole) and then eluted using a gradient of Buffer B (20 mM HEPES-KOH, pH 7.5, 200 mM KCl, 5 mM MgCl$_2$, 300 mM imidazole). NSs::6xHIS was concentrated using a 10 kDa MWCO spin concentrator (Amicon) and further purified by size exclusion chromatography over a Superdex 200 Increase 10/300 GL column (GE Healthcare) in Elution Buffer (20 mM HEPES, 7.5, 200 mM KCl, 5 mM MgCl$_2$, 1 mM TCEP, and 5% Glycerol). The resulting fractions were pooled and flash frozen in liquid nitrogen.

**In vitro NSs/eIF2α-P immunoprecipitation.** Varying combinations of purified eIF2α-P, NSs::6xHIS, eIF2B(αβγδε)$_2$, eIF2Bβγδε, and eIF2Bα$_2$ were incubated (with gentle rocking) with anti-protein C antibody-conjugated resin (generous gift from Aashish Manglik) in Assay Buffer (20 mM HEPES-KOH, pH 7.5, 150 mM KCl, 5 mM MgCl$_2$, 1 mM TCEP, 1 mg/ml bovine serum albumin (BSA), 5 mM CaCl$_2$). After 1.5 h the resin was pelleted by benchtop centrifugation and the supernatant was removed. Resin was washed 3× with 1 ml of ice cold Assay Buffer before resin was resuspended in Elution Buffer (Assay Buffer with 5 mM EDTA and 0.5 mg/ml protein C peptide added) and incubated with gentle rocking for 1 h. The resin was then pelleted and the supernatant was removed. Samples were analyzed by Western Blotting as described above.

**GDP exchange assay.** in vitro detection of BODIPY-FL-GDP binding to eIF2 was performed as previously described[5,8]. The only modification was addition of NSs in certain conditions as indicated. In brief, purified eIF2 (100 nM) was incubated with 100 nM BODIPY-FL-GDP (Thermo Fisher Scientific) in assay buffer (20 mM HEPES-KOH, pH 7.5, 100 mM KCl, 5 mM MgCl$_2$, 1 mM TCEP, and 1 mg/ml BSA) to a volume of 18 μl in 384 square-well black-walled, clear-bottom polystyrene assay plates (Corning). The GEF mix was prepared by incubating a 10× solution of eIF2B(αβγδε)$_2$ with or without 10× solutions of eIF2α-P and/or NSs. To compare nucleotide exchange rates, the 10× GEF mixes were spiked into the 384-well plate wells with a multi-channel pipette, such that the resulting final concentration of eIF2B(αβγδε)$_2$ was 10 nM and the final concentration of other proteins and drugs are as indicated in the figures. Fluorescence intensity was recorded every 10 s for 30–60 min using a Clariostar PLUS (BMG LabTech) plate reader (excitation wavelength: 497 nm, bandwidth 14 nm, emission wavelength: 525 nm, bandwidth: 30 nm). Data were fit to a first-order exponential and plotted in GraphPad Prism 8.

**FAM-ISRIB binding assay.** All fluorescence polarization measurements were performed as previously described[5]. In brief, 20 μl reactions were set up with

100 nM eIF2B(αβγδε)₂ + 2.5 nM FAM-ISRIB (Praxis Bioresearch) in FP buffer (20 mM HEPES-KOH pH 7.5, 100 mM KCl, 5 mM MgCl₂, 1 mM TCEP) and measured in 384-well non-stick black plates (Corning 3820) using the ClarioStar PLUS (BMG LabTech) at room temperature. Prior to reaction setup, eIF2B(αβγδε)₂ was assembled in FP buffer using eIF2Bβγδε and eIF2Bα₂ in 2:1 molar ratio for 1 h at room temperature. FAM-ISRIB was first diluted to 2.5 μM in 100% NMP prior to dilution to 50 nM in 2% NMP and then added to the reaction. For titrations with NSs, dilutions were again made in FP buffer, and the reactions with eIF2B, FAM-ISRIB, and these dilutions ±eIF2α-P were incubated at 22 °C for 30 min prior to measurement of parallel and perpendicular intensities (excitation: 482 nm, emission: 530 nm). Data were plotted in GraphPad Prism 8, and where appropriate, curves were fit to log[inhibitor] vs response function with variable slope.

**Affinity determination and competition analysis by SPR.** NSs affinity determination experiments were performed on a Biacore T200 instrument (Cytiva Life Sciences) by capturing the biotinylated eIF2B(αβγδε)₂, eIF2Bβγδε, and eIF2Bα₂ at ~100 nM on a Biotin CAPture Series S sensor chip (Cytiva Life Sciences) to achieve maximum response ($R_{max}$) of <100 response units (RUs) upon NSs binding. A molar equivalent of each eIF2B species was immobilized. Twofold serial dilutions of purified NSs were flowed over the captured eIF2B complexes at 30 μL/min for 90 s followed by 600 s of dissociation flow. Following each cycle, the chip surface was regenerated with 3 M guanidine hydrochloride. A running buffer of 20 mM HEPES-KOH, pH 7.5, 100 mM KCl, 5 mM MgCl₂, and 1 mM TCEP was used throughout. The resulting sensorgrams were fit to a 1:1 Langmuir binding model using the association and then dissociation model in GraphPad Prism 8.0.

For NSs and eIF2/eIF2-P competition experiments, eIF2B(αβγδε)₂ was immobilized as described above. A solution containing 500 nM NSs, 125 nM eIF2, or 125 nM eIF2-P was flowed over the captured eIF2B for 60 s at 30 μl/min to achieve saturation. Following this binding reaction, a second injection of 500 nM NSs and either 125 nM eIF2 or 125 nM eIF2-P was performed.

**Sample preparation for cryo-electron microscopy.** Decameric eIF2B(αβγδε)₂ was prepared by incubating 20 μM eIF2Bβγδε with 11 μM eIF2Bα₂ in a final solution containing 20 mM HEPES-KOH, pH 7.5, 200 mM KCl, 5 mM MgCl₂, and 1 mM TCEP. This 10 μM eIF2B(αβγδε)₂ sample was further diluted to 750 nM and incubated with 2.25 μM NSs::6xHIS on ice for 1 h before plunge freezing. A 3 μl aliquot of the sample was applied onto the Quantifoil R 1.2/1/3 400 mesh Gold grid and we waited for 30 s. A 0.5 μl aliquot of 0.1–0.2% Nonidet P-40 substitute was added immediately before blotting. The entire blotting procedure was performed using Vitrobot (FEI) at 10 °C and 100% humidity.

**Electron microscopy data collection.** Cryo-EM data for the eIF2B–NSs complex were collected on a Titan Krios transmission electron microscope operating at 300 keV, and micrographs were acquired using a Gatan K3 direct electron detector. Serial EM was used to collect the EM data[53]. The total dose was $67e^-/\text{Å}^2$, and 117 frames were recorded during a 5.9 s exposure. Data wereas collected at ×105,000 nominal magnification (0.835 Å/pixel at the specimen level), and nominal defocus range of −0.6 to −2.0 μm.

**Image processing.** The micrograph frames were aligned using MotionCorr2[54]. The contrast transfer function (CTF) parameters were estimated with GCTF[55]. Particles were picked in Cryosparc v2.15 using the apo eIF2B (EMDB: 23209) as a template. Particles were extracted using a 80-pixel box size[56], and classified in 2D[57]. Classes that showed clear protein features were selected and extracted for ab initio reconstruction followed by homogeneous and heterogeneous refinement. Particles belonging to the best class were then re-extracted with a pixel size of 2.09 Å, and then subjected to nonuniform refinement, yielding a reconstruction of 4.25 Å. These particles were subjected to another round of heterogeneous refinement followed by nonuniform refinement to generate a consensus reconstruction consisting of the best particles. These particles were re-extracted at a pixel size of 0.835 Å. Then, CTF refinement was performed to correct for the per-particle CTF as well as beam tilt. A final round of 2D classification followed by nonuniform refinement was performed to yield the final structure of 2.6 Å.

**Atomic model building, refinement, and visualization.** To build models for the eIF2B–NSs complex, the previously determined structures of the human eIF2B in its apo form (PDB 7L70) was used as the starting model for the eIF2B part[5]. To build the NSs model, we first ran the structure prediction program RaptorX using the full-length NSs sequence[58]. The predicted structure is divided into two parts: the C-terminal domain predicted based on the structure of the RVFV NSs (PDB 5OOO) and the N-terminal domain is predicted without a known PDB structure as a template[31]. The predicted full-length structure was docked into the EM density corresponding to the NSs in UCSF Chimera[59], and then subjected to rigid body refinement in Phenix[60]. The models were then manually adjusted in Coot[61] and then refined in phenix.real_space_refine[60] using global minimization, secondary structure restraints, Ramachandran restraints, and local grid search. Then iterative cycles of manual rebuilding in Coot and phenix.real_space_refine were performed. The final model statistics were tabulated using Molprobity[62]. Distances were calculated from the atomic models using UCSF Chimera. Molecular graphics and

analyses were performed with the UCSF Chimera package[59]. UCSF Chimera is developed by the Resource for Biocomputing, Visualization, and Informatics and supported by NIGMS P41-GM103311. The atomic model is deposited in the PDB under accession code 7RLO. The EM map is deposited into EMDB under accession code EMD-24535.

**Reporting summary.** Further information on research design is available in the Nature Research Reporting Summary linked to this article.

## Data availability

The data that support this study are available from the corresponding author upon reasonable request. The cryo-EM structure generated in this study has been deposited in the protein data bank under the accsssion code 7RLO. The corresponding EM map has been deposited in the EM database under the accession code EMD-24535. The structure of the RVFV NSs used for model building is available in the protein data bank under the accession code 5OOO. Source data are provided with this paper.

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

## Acknowledgements

We thank the Walter lab for helpful discussions throughout the course of this project; Cobar Wollemi for inspiring Supplementary Fig. 10; Calico Life Sciences LLC for a generous gift of purified eIF2 heterotrimer; Z. Yu and D. Bulkley of the UCSF Center for Advanced Cryo-EM facility, which is supported by NIH grants S10OD021741 and S10OD020054; and the Howard Hughes Medical Institute (HHMI). We also thank the QB3 shared cluster for computational support. This work was supported by generous support from Calico Life Sciences LLC (to P.W.); a generous gift from The George and Judy Marcus Family Foundation (to P.W.); the Belgian-American Educational Foundation (BAEF) Postdoctoral Fellowship (to M.B.), the Damon Runyon Cancer Research Foundation Postdoctoral fellowship (to L.W.); and the Jane Coffin Child Foundation Postdoctoral Fellowship (to R.L.). A.F. is a HHMI faculty scholar and a Chan Zuckerberg Biohub investigator. P.W. is an Investigator of the Howard Hughes Medical Institute.

## Author contributions

P.W. supervised the research. M.S., L.W., J.Z.C., and R.E.L. designed the experiments. M.S. performed all cloning. M.S., J.Z.C., R.E.L., and M.B. expressed and purified proteins. M.S. and J.Z.C. generated the cell lines, and performed the flow cytometry experiments. M.S. performed the binding assays (SPR and bead immobilization). M.S., J.Z.C., and R.E.L. performed the nucleotide exchange assays. M.S., J.Z.C., and M.B. performed the FAM-ISRIB-binding assays. M.S. performed all western blotting. L.W. performed cryo-EM sample preparation, data collection, processing, and model building with A.F. providing additional model building input. M.S., L.W., and P.W., prepared the rough manuscript draft, with finalizing input from all authors including J.Z.C., R.E.L., M.B., J.D.W., and A.F.

## Competing interests

P.W. is an inventor on U.S. Patent 9708247 held by the Regents of the University of California that describes ISRIB and its analogs. Rights to the invention have been licensed by UCSF to Calico. The remaining authors declare no competing interests.
