## [Peer Review File · Nature Communications]

Viral Evasion of the Integrated Stress Response Through Antagonism of eIF2-P binding to eIF2BReviewers' Comments:

Reviewer #1:

Remarks to the Author:

Schoof et al present a structure-function analysis that helps to explain how Sandfly Fever Sicilian virus (SFSV) protein NSs antagonises the integrated stress response by targeting the central player eIF2B. The main experimental approaches used are CryoEM where the authors present a 2.6 Å structure of NSs bound to eIF2B. This demonstrates NSs binds to the place where phosphorylated eIF2 was previously shown to bind between eIF2B α and δ . Unlike eIF2, NSs only makes extensive contact with eIF2B α . This structural work is accompanied by both cell (K562 cells) and in vitro analyses, which show that NSs can attenuate activity from 3 of 4 eIF2 kinases, although the impact on global translation repression through GCN2 was not impacted, probably because Gcn2 is activated by translation elongation pauses that also reduce global translation independently of ISR control. Mutagenesis of surface residues of NSs implicated in binding eIF2B have appropriate loss of activity. No mutagenesis of the opposite side of the interface (eIF2B α) has been done, but there are likely few residues that could be targeted without also impacting of eIF2(P) binding, but this point is unclear from the presentation. It would be of interest to show the surface contact areas of eIF2 α and NSs on eIF2B α , so what is shared and what is distinct is made clear.

In general the work is of very high standard. My major query is with the measured affinity between NSs and eIF2B that does not appear tight enough for the observed inhibitory action of NSs enough when compared with previous es

Major points.

1. The in vitro binding kinetics appear to show a modest affinity 43 nM here when compared with previous determinates of eIF2(P) affinity for eIF2B of 0.3-3 nM, see [Bogorad, A.M., Lin, K.Y. & Marintchev, A. eIF2B Mechanisms of Action and Regulation: A Thermodynamic View. *Biochemistry* 57, 1426-1435 (2018).] for a recent comprehensive review of the topic. As these estimates suggest NSs is 10-100 x poorer in binding to eIF2B, this affinity estimate is hard to reconcile with the remainder of the data presented, unless NSs expression is several orders of magnitude above that of eIF2. I note that this K_d value was obtained using only three different concentrations of ligand using SPR with eIF2B (the larger partner) immobilized. The kinetic determination requires a larger range of concentrations.

In addition, what stoichiometry is assumed here, as one eIF2B binds 2 NSs in the cryoEM?

Would performing the experiments with immobilized NSs give the same affinity?

What is the affinity of eIF2B for the full phosphorylated eIF2 complex in these assays?

2. Isolated eIF2 α , used in Figure 3b has a lower affinity for eIF2B than the full complex, so is not a fair competitor. In vivo NSs will have to compete with full eIF2 complexes which the authors use in their GEF assays. It would be helpful to have a fair competitive assay that mimics the proteins in the cell.

3. Is the eIF2B conformation found with NSs bound compatible with eIF2 binding between eIF2B β - δ , or is there any steric clash if the authors previous 'a-state' structure is superimposed on the new structure here? the authors have not presented a structure for eIF2 and NSs bound to eIF2B at the same time.

minor points.

'mimicry' is not used appropriate in the title as the structure of NSs does not demonstrate macromolecular mimicry of eIF2 α . It demonstrates an overlapping binding site, but as the authors' study shows, it does not mimic structure or the binding interface of eIF2 α with eIF2B. A viral regulator that is a mimic is the Vaccinia virus kinase inhibitor K3L that is a mimic of eIF2 α NTD.

The statements within lines 60-67 require referencing.

line 80 punctuation error

line 126-7. The last clause about ISRIB and a potential compromised eIF2B complex state formation is not appropriate or necessary here.

The alignment in extended data 4 would be more informative if secondary structure assignments for each protein could be added.

line 241. Should F5 be Y5 or F7?

line 394 No NSs in the last complex in panel a) I think.

Reviewer #2:

Remarks to the Author:

Activation of the protein kinase PKR by viral dsRNAs plays a critical role in the innate immune response. Phosphorylation of the translation factor eIF2 by PKR converts eIF2 from a substrate to an inhibitor of its guanine nucleotide exchange factor eIF2B, and thereby inhibits translation. The inhibition of cellular and viral mRNA translation is expected to inhibit viral replication. To compete against this host defense, viruses have evolved mechanisms to subvert PKR and the inhibition of translation by phosphorylated eIF2. In this paper, the authors examine a recently described inhibitor from Sandfly Fever Sicilian virus (SFSV). The NSs protein from SFSV was recently proposed to thwart the PKR-directed antiviral response by binding to eIF2B and rendering it insensitive to inhibition by phosphorylated eIF2. In this work, the authors demonstrate that NSs binds to eIF2B and competitively inhibits the inhibitory binding of phosphorylated eIF2. Consistent with NSs targeting eIF2B, the authors show that NSs also suppresses the translational inhibitory properties of other eIF2 kinases in addition to PKR. Using cryo-EM, the authors show that NSs binds to eIF2B in a site overlapping with the site occupied by phosphorylated eIF2, thus providing a clear rationale for the mechanism of NSs rescuing eIF2B from inhibition by phosphorylated eIF2. Finally, by mutating residues predicted to be critical for the binding of NSs to eIF2B, the authors provide independent biochemical support for the structure.

Taken together, this is an exciting paper that provides new insights into a novel mechanism of viral evasion of host cell defenses. The paper is well written, the data are of high quality, and the story is both convincing and of general interest.

I have a few minor comments that the authors should address:

1. Figure 1. It would be nice to examine eIF2 α phosphorylation in these experiments to confirm that NSs does alter eIF2 phosphorylation levels in cells with activated PERK, HRI or GCN2. The data in Fig. 6 show that NSs does not reduce eIF2 phosphorylation in cells with activated PERK, but it might be nice to show this for HRI and GCN2 as well

2. Line 191: it might be better to cite Fig. 4A rather than Extended Data Fig. 3 to show NSs binding at two sites on eIF2B

3. Line 241: Should F5 be F7?

4. Line 261: I don't think that "binding" is the proper word here. The data in Figure 6 examine ATF4 expression and general translation as indirect measures of eIF2B activity. While it is possible (perhaps likely) that the mutations are affecting the K_d of NSs binding to eIF2B, alternatively the mutations might alter the position of NSs on eIF2B such that eIF2B is now susceptible to inhibition by phosphorylated eIF2. It is probably better to say that the aromatic fingers appear to be critical for NSs protection of eIF2B from the inhibitory effects of phosphorylated eIF2.

Line 269 likewise discusses NSs binding based on data examining ATF4 and general translation. As the statement on line 269 is qualified by "suggests" it might be okay.

This same idea is also relevant to "binding" in line 325.

Reviewer #3:

Remarks to the Author:

The initiation factor eIF2 brings the initiator Met-tRNA_i to the ribosome and is one of the main targets of regulation of protein synthesis. Upon certain stress conditions that lead to the activation of the integrated stress response (ISR), eIF2 is phosphorylated on a single serine and acts as an inhibitor instead of as a substrate of the decameric eIF2B protein. eIF2B is the specific guanine nucleotide exchange factor (GEF) of eIF2. One of these stress signals is caused by viral infections that trigger the activation of the PKR kinase, and that activation results in the phosphorylation of eIF2. The phosphorylation of only a small fraction of eIF2 can cause a substantial decrease in protein synthesis because eIF2 is much more abundant than eIF2B in cells

Several mechanisms of viral evasion of ISR have been described and target almost every step of the ISR. Here the authors provide functional and structural insights on how the Sandfly Fever Sicilian Virus (SFSV) inhibits ISR by releasing eIF2B from its unproductive complex with phosphorylated eIF2.

Overall, this is a strong and well-written manuscript that will move the field of eIF2B regulation forward. This field is a subject of high interest by several labs; in fact this manuscript is very similar in its content and main conclusions to a recent paper deposited in BioRxiv by the Ito lab.

Some thoughts on how the paper could be improved are described below. In more detail, I have only one main concern and a few minor concerns:

Major concern:

1) The authors used a combination of cellular, biochemical and structural experiments that I think are appropriate, and with such data in mind I really believe that SFSV evades ISR by competitively blocking the eIF2B-eIF2-P interaction. However to be completely sure of that, the NSs mutants produced to functionally validate the cryoEM structure, should be used not only in cellular assays but also in in vitro interaction assays (SPR assays) and also in GEF activity assays. In fact in the absence of such assays I disagree with the conclusions raised on sentences found in page 10, lines 267-273, since cellular assays do not provide a direct proof (or lack of) of interaction. The same can be said to the sentence "... changing them disrupts binding" found in line 325. Moreover it would be very interesting to see the impact of the D37A mutant in eIF2B δ on complex formation, monitored in SPR assays; I would expect a highly detrimental effect of that mutant on eIF2B-NSs complex formation given the total absence of interaction between NSs and eIF2B α dimers, as shown in figure 2f.

Minor comments

1) You have found that the K_d of the eIF2B-NSs complex formation measured by SPR is 43nM.

Previous work using other techniques has demonstrated that the interaction of eIF2B with eIF2-P is also in the low nanomolar range, but a fair comparison using the same method/conditions would be desirable.

2) You do not see any improvement in the GEF activity of eIF2B when complexed with NSs, in the absence of phosphorylated eIF2. Although this observation is in agreement with the conformational similarity of eIF2B found in the cryoEM structures of eIF2B in its apo form, the active eIF2B in complex with eIF2 and eIF2B in complex with either NSs or ISRIB, I would (at least) expect a slight improvement in the activity of eIF2B when bound to a molecule/protein that acts stabilizing its active conformation. Can you comment on that?

3) In your GEF assays, you use an eIF2 protein that has been expressed in HEK cells, and therefore eIF2 could be partially phosphorylated. Have you checked (for example by mass spectrometry) that

your sample is 100% unphosphorylated?

4) CryoEM structure refinement: You did not apply any symmetry but it seems (based on the RMSD) that the two halves of the structure are quite similar. Although the overall resolution is already really good, have you considered refining the map using C2 symmetry to boost the resolution?

5) H36 of NSs interacts with R321 of eIF2B δ . It does not seem a very favourable interaction. What is the pH of the buffer used for cryoEM grid preparation? (it is not specified in material and methods). May H36 mainly contribute to the eIF2B-eIF2 complex formation indirectly through an internal interaction with D37, as seen in extended figure S7?

6) How conserved across different species are the NSs-interacting residues of eIF2B. Based on that, could SFSV regulate ISR in other mammals/eukaryotes?

7) In a recent paper (Zyryanova et al 2021 Mol Cell), the authors showed that a decamer of eIF2B can interact with only one heterotrimer of eIF2, in contrast with the more symmetrical cryoEM structures of the eIF2B-eIF2-P complex published earlier. This was accomplished after extensive classification in different classes during cryoEM processing. Are you sure you did not miss such a class (eIF2B decamer with only one monomer of NSs) during cryoEM data processing? In extended figure 3b, you show two rounds of heterogeneous refinement in CryoSparc. Have you further classified the discarded classes coming out from these two rounds of heterogeneous refinement?

8) Page 12, Lines 322-323: I do not really understand what do you mean by "cooperative". I would use another word, since "cooperativity" has a completely different meaning in oligomeric proteins/enzymes and you do not prove that the binding of 1 molecule of NSs to one half of eIF2B has an effect on the binding of a second molecule of NSs to eIF2B. Moreover, any kind of interaction involving different regions of a protein can be considered "cooperative" in the way you use that word. Change "cooperativity" also in figure legend of extended figure 7.

9) Figures 4 and 5: You should use slightly different colours for the various eIF2B subunits, perhaps different shades of blue. This is especially important for panels showing eIF2B α and eIF2B δ simultaneously, for example figure 4d, 5a, 5b. Add a label in every residue shown in figure 5b. Labels in Fig 5e are difficult to follow unless you change to blue the labels of residues belonging to eIF2B.

10) Extended figure 3b. Line above second "heterogeneous refinement" is confusing. Please remove it.

11) Extended table 1. Please specify the percentiles at a given resolution that Molprobity provides.

12) Section describing the sample preparation of grids for cryoEM. I do not really understand why the eIF2B decamer is at 10 μ M after mixing 20 μ M octamers with 11 μ M eIF2B α dimers

13) Lines 715-718. When describing the two parts of structure predicted by RaptorX, I think you mixed up N-terminal and C-terminal domains.

Detailed Responses to the Reviewers' Comments:

Reviewer #1 (Remarks to the Author):

Schoof et al present a structure-function analysis that helps to explain how Sandfly Fever Sicilian virus (SFSV) protein NSs antagonises the integrated stress response by targeting the central player eIF2B. The main experimental approaches used are CryoEM where the authors present a 2.6 Å structure of NSs bound to eIF2B. This demonstrates NSs binds to the place where phosphorylated eIF2 was previously shown to bind between eIF2Balpha and delta. Unlike eIF2, NSs only makes extensive contact with eIF2Balpha. This structural work is accompanied by both cell (K562 cells) and in vitro analyses, which show that NSs can attenuate activity from 3 of 4 eIF2 kinases, although the impact on global translation repression through GCN2 was not impacted, probably because Gcn2 is activated by translation elongation pauses that also reduce global translation independently of ISR control. Mutagenesis of surface residues of NSs implicated in binding eIF2B have appropriate loss of activity. No mutagenesis of the opposite side of the interface (eIF2Balpha) has been done, but there are likely few residues that could be targeted without also impacting of eIF2(P) binding, but this point is unclear from the presentation. It would be of interest to show the surface contact areas of eIF2alpha and NSs on eIF2Balpha, so what is shared and what is distinct is made clear.

In general the work is of very high standard. My major query is with the measured affinity between NSs and eIF2B that does not appear tight enough for the observed inhibitory action of NSs enough when compared with previous es

Major points.

1. The in vitro binding kinetics appear to show a modest affinity 43 nM here when compared with previous determinates of eIF2(P) affinity for eIF2B of 0.3-3 nM, see [Bogorad, A.M., Lin, K.Y. & Marintchev, A. eIF2B Mechanisms of Action and Regulation: A Thermodynamic View. *Biochemistry* 57, 1426-1435 (2018).] for a recent comprehensive review of the topic. As these estimates suggest NSs is 10-100 x poorer in binding to eIF2B, this affinity estimate is hard to reconcile with the remainder of the data presented, unless NSs expression is several orders of magnitude above that of eIF2. I note that this Kd value was obtained using only three different concentrations of ligand using SPR with eIF2B (the larger partner) immobilized. The kinetic determination requires a larger range of concentrations.

In addition, what stoichiometry is assumed here, as one eIF2B binds 2 NSs in the cryoEM?

Would performing the experiments with immobilized NSs give the same affinity?

What is the affinity of eIF2B for the full phosphorylated eIF2 complex in these assays?

We thank the reviewer for raising many nuanced points about competition between eIF2-P and NSs. We have undertaken a number of new experiments to address these concerns. Regarding the concentrations used to determine the NSs affinity for eIF2B, we note that in the reviewed version of the manuscript

there were 4 concentrations not 3 as the reviewer stated. Nevertheless, we have updated the SPR experiment to include 5 concentrations, with a calculated K_D of 30 nM. A closely matching affinity was independently determined by Takuhiro Ito's group (17 nM) using a different technique where eIF2B concentration was titrated (Kashiwagi et al., 2021). Therefore, we have high confidence in our measurement.

Further, we have added data to show that the affinity of FAM-ISRIB binding to decameric eIF2B (~24 nM) is comparable to the NSs affinity, showing that molecules in this affinity range can certainly impact the ISR through eIF2-P antagonism.

Additionally, we have undertaken co-binding assays using full heterotrimeric eIF2 and eIF2-P (fig. 3d-g) and show that NSs and eIF2 can co-bind but NSs and eIF2-P cannot co-bind.

Taken together, we do not see any "hard to reconcile" data. NSs and ISRIB both bind to eIF2B with independently verified low nanomolar affinities and are capable of antagonizing eIF2-P binding in many different assays. Usually, viral proteins are expressed far in excess over host protein levels, and high levels of NSs relative to eIF2-P are entirely reasonable to expect during a viral infection. Combining our new data with our multiple binding assays and a 2.6 Å cryo-EM structure, there is no doubt that NSs binding to eIF2B directly blocks eIF2-P binding.

2. Isolated eIF2 α , used in Figure 3b has a lower affinity for eIF2B than the full complex, so is not a fair competitor. In vivo NSs will have to compete with full eIF2 complexes which the authors use in their GEF assays. It would be helpful to have a fair competitive assay that mimics the proteins in the cell.

We have added an SPR experiment showing that NSs can block eIF2-P heterotrimer binding. We will also note that the GEF assay the reviewer refers to uses the phosphorylated alpha subunit alone, not the full heterotrimer. While eIF2-P certainly binds eIF2B tighter than eIF2 α -P, from a mechanistic standpoint the use of eIF2 α -P is entirely fair as it is the minimal unit responsible for inhibition of eIF2B.

3. Is the eIF2B conformation found with NSs bound compatible with eIF2 binding between eIF2B beta-delta, or is there any steric clash if the authors previous 'a-state' structure is superimposed on the new structure here? the authors have not presented a structure for eIF2 and NSs bound to eIF2B at the same time.

We have added an SPR experiment showing that NSs and eIF2 can co-bind. We also note that eIF2B is in the A-State in the apo-structure, in the ISRIB-bound

structure, in the eIF2-bound structure, and in the NSs-bound structure, and the conformation of eIF2B in these complexes are nearly identical. Also, if eIF2 would not bind to the NSs-occupied eIF2B, then NSs would be scored as an inhibitor, and we would expect to see a decrease in GEF activity in extended data fig. 2, which we do not.

minor points.

'mimicry' is not used appropriate in the title as the structure of NSs does not demonstrate macromolecular mimicry of eIF2alpha. It demonstrates an overlapping binding site, but as the authors' study shows, it does not mimic structure or the binding interface of eIF2alpha with eIF2B. A viral regulator that is a mimic is the Vaccinia virus kinase inhibitor K3L that is a mimic of eIF2alpha NTD.

This point is well taken – we have updated the title.

The statements within lines 60-67 require referencing.

We have updated referencing in this region and throughout to ensure correct citations.

line 80 punctuation error

We fixed the error.

line 126-7. The last clause about ISRIB and a potential compromised eIF2B complex state formation is not appropriate or necessary here.

We have reviewed this clause and think it is an appropriate introduction to ISRIB as a modulator of the ISR.

The alignment in extended data 4 would be more informative if secondary structure assignments for each protein could be added.

We have updated this figure to include the suggested data.

line 241. Should F5 be Y5 or F7?

It should be F7, and we fixed this error. Thank you for spotting this.

line 394 No NSs in the last complex in panel a) I think.

We fixed this issue.

Reviewer #2 (Remarks to the Author):

Activation of the protein kinase PKR by viral dsRNAs plays a critical role in the innate immune response. Phosphorylation of the translation factor eIF2 by PKR converts eIF2 from a substrate to an inhibitor of its guanine nucleotide exchange factor eIF2B, and thereby inhibits translation. The inhibition of cellular and viral mRNA translation is expected to inhibit viral replication. To compete against this host defense, viruses have evolved mechanisms to subvert PKR and the inhibition of translation by phosphorylated eIF2. In this paper, the authors examine a recently described inhibitor from Sandfly Fever Sicilian virus (SFSV). The NSs protein from SFSV was recently proposed to thwart the PKR-directed antiviral response by binding to eIF2B and rendering it insensitive to inhibition by phosphorylated eIF2. In this work, the authors demonstrate that NSs binds to eIF2B and competitively inhibits the inhibitory binding of phosphorylated eIF2. Consistent with NSs targeting eIF2B, the authors show that NSs also suppresses the translational inhibitory properties of other eIF2 kinases in addition to PKR. Using cryo-EM, the authors show that NSs binds to eIF2B in a site overlapping with the site occupied by phosphorylated eIF2, thus providing a clear rationale for the mechanism of NSs rescuing eIF2B from inhibition by phosphorylated eIF2. Finally, by mutating residues predicted to be critical for the binding of NSs to eIF2B, the authors provide independent biochemical support for the structure.

Taken together, this is an exciting paper that provides new insights into a novel mechanism of viral evasion of host cell defenses. The paper is well written, the data are of high quality, and the story is both convincing and of general interest.

I have a few minor comments that the authors should address:

1. Figure 1. It would be nice to examine eIF2 α phosphorylation in these experiments to confirm that NSs does alter eIF2 phosphorylation levels in cells with activated PERK, HRI or GCN2. The data in Fig. 6 show that NSs does not reduce eIF2 phosphorylation in cells with activated PERK, but it might be nice to show this for HRI and GCN2 as well

While we agree that such data could be added, we do not think that they would sufficiently firm-up or enhance the conclusions of the manuscript to warrant their inclusion. As the reviewer notes, in fig. 6 we show that NSs does not lead to a reduction in eIF2 phosphorylation after PERK activation, and, combined with the prior data from Wuerth et al., 2020, there is no reason to think that it would be different after activation of any of the other kinases.

2. Line 191: it might be better to cite Fig. 4A rather than Extended Data Fig. 3 to show NSs binding at two sites on eIF2B

We have updated the sentence and elected to cite both figure panels

3. Line 241: Should F5 be F7?

Yes, and we corrected this mistake. Thank you for spotting this one.

4. Line 261: I don't think that "binding" is the proper word here. The data in Figure 6 examine ATF4 expression and general translation as indirect measures of eIF2B activity. While it is possible (perhaps likely) that the mutations are affecting the K_d of NSs binding to eIF2B, alternatively the mutations might alter the position of NSs on eIF2B such that eIF2B is now susceptible to inhibition by phosphorylated eIF2. It is probably better to say that the aromatic fingers appear to be critical for NSs protection of eIF2B from the inhibitory effects of phosphorylated eIF2.

Line 269 likewise discusses NSs binding based on data examining ATF4 and general translation. As the statement on line 269 is qualified by "suggests" it might be okay.

This same idea is also relevant to "binding" in line 325.

We have updated the language to be more in accord with the cellular data.

Reviewer #3 (Remarks to the Author):

The initiation factor eIF2 brings the initiator Met-tRNA_i to the ribosome and is one of the main targets of regulation of protein synthesis. Upon certain stress conditions that lead to the activation of the integrated stress response (ISR), eIF2 is phosphorylated on a single serine and acts as an inhibitor instead of as a substrate of the decameric eIF2B protein. eIF2B is the specific guanine nucleotide exchange factor (GEF) of eIF2. One of these stress signals is caused by viral infections that trigger the activation of the PKR kinase, and that activation results in the phosphorylation of eIF2. The phosphorylation of only a small fraction of eIF2 can cause a substantial decrease in protein synthesis because eIF2 is much more abundant than eIF2B in cells. Several mechanisms of viral evasion of ISR have been described and target almost every step of the ISR. Here the authors provide functional and structural insights on how the Sandfly Fever Sicilian Virus (SFSV) inhibits ISR by releasing eIF2B from its unproductive complex with phosphorylated eIF2.

Overall, this is a strong and well-written manuscript that will move the field of eIF2B regulation forward. This field is a subject of high interest by several labs; in fact this manuscript is very similar in its content and main conclusions to a recent paper deposited in BioRxiv by the Ito lab.

Some thoughts on how the paper could be improved are described below. In more detail, I have only one main concern and a few minor concerns:

Major concern:

1) The authors used a combination of cellular, biochemical and structural experiments that I think are appropriate, and with such data in mind I really believe that SFSV evades ISR by competitively blocking the eIF2B-eIF2-P interaction. However to be

completely sure of that, the NSs mutants produced to functionally validate the cryoEM structure, should be used not only in cellular assays but also in in vitro interaction assays (SPR assays) and also in GEF activity assays. In fact in the absence of such assays I disagree with the conclusions raised on sentences found in page 10, lines 267-273, since cellular assays do not provide a direct proof (or lack of) of interaction. The same can be said to the sentence "... changing them disrupts binding" found in line 325. Moreover it would be very interesting to see the impact of the D37A mutant in eIF2B δ on complex formation, monitored in SPR assays; I would expect a highly detrimental effect of that mutant on eIF2B-NSs complex formation given the total absence of interaction between NSs and eIF2B α dimers, as shown in figure 2f.

While we agree that the purification of point mutants could be of interest, we do not think that it would enhance any of the critical conclusions in the manuscript. As the reviewer states him/herself "SFSV evades ISR by competitively blocking the eIF2B-eIF2-P interaction". Further, Takuhiro Ito independently generated point mutants of the aromatic fingers and showed a major impact on binding to eIF2B (Kashiwagi et al., 2021). In combination, Ito's in vitro data and our in vivo data convincingly support the importance of the aromatic fingers. We have updated our manuscript to reference this independent work which is currently published in bioRxiv. We have also updated the language to be more in line with the cellular nature of the assays.

Minor comments:

1) You have found that the Kd of the eIF2B-NSs complex formation measured by SPR is 43nM. Previous work using other techniques has demonstrated that the interaction of eIF2B with eIF2-P is also in the low nanomolar range, but a fair comparison using the same method/conditions would be desirable.

Please see above (Reviewer 1, major comment 1) discussion on this topic.

2) You do not see any improvement in the GEF activity of eIF2B when complexed with NSs, in the absence of phosphorylated eIF2. Although this observation is in agreement with the conformational similarity of eIF2B found in the cryoEM structures of eIF2B in its apo form, the active eIF2B in complex with eIF2 and eIF2B in complex with either NSs or ISRIB, I would (at least) expect a slight improvement in the activity of eIF2B when bound to a molecule/protein that acts stabilizing its active conformation. Can you comment on that?

This is a very good point. It is certainly possible in principle that NSs (or, indeed, ISRIB) could improve eIF2B's GEF activity. We have done such experiments; however, if there is an improvement it is so minor that we cannot confidently say so because it is too small to stand out from the noise. From this and our

structural experiments, we conclude that the A-State / I-State equilibrium of eIF2B largely populates the A-State, unless perturbed by eIF2-P binding.

3) In your GEF assays, you use an eIF2 protein that has been expressed in HEK cells, and therefore eIF2 could be partially phosphorylated. Have you checked (for example by mass spectrometry) that your sample is 100% unphosphorylated?

We have checked the protein by running it on a Phos-tag gel (which separates cleanly by phosphorylation state) and confirmed that eIF2 is fully unphosphorylated

4) CryoEM structure refinement: You did not apply any symmetry but it seems (based on the RMSD) that the two halves of the structure are quite similar. Although the overall resolution is already really good, have you considered refining the map using C2 symmetry to boost the resolution?

Thank you for your suggestion. We tried refining by applying C2 symmetry and the resolution improved from 2.6 Å to 2.5 Å. In other words, processing with symmetry only marginally changed the overall resolution without qualitatively improving the map, thus has no meaningful impact on the point of the paper. We therefore kept the current refinement strategy.

5) H36 of NSs interacts with R321 of eIF2B δ . It does not seem a very favourable interaction. What is the pH of the buffer used for cryoEM grid preparation? (it is not specified in material and methods). May H36 mainly contribute to the eIF2B-eIF2 complex formation indirectly through an internal interaction with D37, as seen in extended figure S7?

The pH of the buffer was 7.5 – we have updated the methods accordingly. The interaction between H36 and R321 is a cation-pi interaction (the histidine provides the aromatic ring and the arginine the positive charge). The reviewer could be right in that the strength of this interaction is not strong enough to be the major contributor to its function, in which case H36 may mainly contribute through the interaction with D37. A sentence regarding this is included in the manuscript.

6) How conserved across different species are the NSs-interacting residues of eIF2B. Based on that, could SFSV regulate ISR in other mammals/eukaryotes?

Below is the sequence alignment of eIF2B α across a few mammalian species (from top to bottom: bovine, human, dog, rat, mouse). The amino acids (highlighted in blue boxes) that are involved in NSs interaction are mostly

and our group have published structures of eIF2B in complex with 2 copies of eIF2. Further, the graphical abstract in the Zyryanova paper shows eIF2B engaging 2 copies of eIF2.

8) Page 12, Lines 322-323: I do not really understand what do you mean by “cooperative”. I would use another word, since “cooperativity” has a completely different meaning in oligomeric proteins/enzymes and you do not prove that the binding of 1 molecule of NSs to one half of eIF2B has an effect on the binding of a second molecule of NSs to eIF2B. Moreover, any kind of interaction involving different regions of a protein can be considered “cooperative” in the way you use that word. Change “cooperativity” also in figure legend of extended figure 7.

We have removed mentions of cooperativity and instead refer to it now as “synergistic”.

9) Figures 4 and 5: You should use slightly different colours for the various eIF2B subunits, perhaps different shades of blue. This is especially important for panels showing eIF2B α and eIF2B δ simultaneously, for example figure 4d, 5a, 5b. Add a label in every residue shown in figure 5b. Labels in Fig 5e are difficult to follow unless you change to blue the labels of residues belonging to eIF2B.

Thank you for bringing this up. We have reviewed the figures and made some modifications that should improve readability, in particular for figure 5 where we tweaked the shades of eIF2B α and eIF2B δ and fixed the labeling issues.

10) Extended figure 3b. Line above second “heterogeneous refinement” is confusing. Please remove it.

We removed the line and updated this figure.

11) Extended table 1. Please specify the percentiles at a given resolution that Molprobity provides.

We have included percentiles and updated Table 1.

12) Section describing the sample preparation of grids for cryoEM. I do not really understand why the eIF2B decamer is at 10 μ M after mixing 20 μ M octamers with 11 μ M eIF2B α dimers

We use 20 μ M tetramers not octamers. 2 tetramers are in 1 decamer, which explains why it ends up as a 10 μ M decamer.

13) Lines 715-718. When describing the two parts of structure predicted by RaptorX, I think you mixed up N-terminal and C-terminal domains.

Thank you for spotting this mistake. We fixed it.

Reviewers' Comments:

Reviewer #1:

Remarks to the Author:

This revised version by Schoof et al, has made several changes to the text and included some additional data to address the reviewers points raised. As the authors indicate some other points are addressed by a competitor manuscript not yet published.

The changes have strengthened what was generally a very good manuscript.

old point

Affinities.

If the authors are not going to assess the eIF2/eIF2-P affinity for eIF2B by SPR here as suggested, then they should acknowledge at an appropriate point in the results (when Fig 2d is discussed) the previously determined measurements, as was raised in my original review and also by reviewer 3.

new points.

The new Figure 3 panels d-g data indicate that there is competition between NSs and eIF2-P. The relative change in RU suggest that the order of ligand binding to eIF2B can impact the outcome.

1. How reproducible are these data, n=1?

2. How do the authors interpret the lack of change in RU in e) when eIF2-P addition follows NSs and vice-versa in g)? Can NSs can only prevent inhibition of eIF2B by eIF2-P if it binds first as eIF2-P:eIF2B is very stable?

3. Does the more modest increase in RU in f) following NSs addition than seen for the reverse binding in d) mean that less NSs is bound to eIF2-eIF2B than to eIF2B alone here?

The descriptions in the results (lines 177-179) explaining these data should be expanded to make the findings clearer.

4. The use of 'synergy' to describe the binding of NSs to eIF2B is not appropriate and adds nothing to the discussion, I suggest removing it. lines 330 and 503.

5. line 268 'independently' typo spotted.

6. line 338 describes eIF2B as having tetrameric halves, when it has pentameric halves. Recent papers from the Walter lab and from the Ron lab show ISRIB does not 'staple' together decameric eIF2B, it is naturally decameric in vivo. ISRIB is primarily an allosteric activator. This sentence needs revising.

Reviewer #3:

Remarks to the Author:

I acknowledge that the authors took into consideration most of my remarks from the first review.

I do not have further concerns at the present reviewing stage.

REVIEWERS' COMMENTS

Reviewer #1 (Remarks to the Author):

This revised version by Schoof et al, has made several changes to the text and included some additional data to address the reviewers points raised. As the authors indicate some other points are addressed by a competitor manuscript not yet published. The changes have strengthened what was generally a very good manuscript.

old point

Affinities.

If the authors are not going to assess the eIF2/eIF2-P affinity for eIF2B by SPR here as suggested, then they should acknowledge at an appropriate point in the results (when Fig 2d is discussed) the previously determined measurements, as was raised in my original review and also by reviewer 3.

A citation to the previously determined affinities has been added to the text.

Evidence for a competition with eIF2-P is not presented until figure 3 (so when 2d is discussed is not an appropriate point). Owing to this and concerns about flow of the text we have put the reference and suggested sentence in the discussion.

new points.

The new Figure 3 panels d-g data indicate that there is competition between NSs and eIF2-P. The relative change in RU suggest that the order of ligand binding to eIF2B can impact the outcome.

1. How reproducible are these data, n=1?

Panels 3d-g are n = 1, as is standard practice with most SPR data for straightforward epitope binning / competition experiments (See Figure 4B in Zheng et. al., 2021 doi.org/10.3389/fimmu.2021.641819; Figure 2B in Sivasubramanian et. al., 2017 10.1080/19420862.2016.1246096; or Figure 1D from our group's recent paper Schoof et. al., 2020 10.1126/science.abe3255). Further, the data presented here and elsewhere in our manuscript all support the same conclusion – that NSs blocks eIF2-P binding while allowing eIF2 binding. We have now presented 4 orthogonal assays (FAM-ISRIB competition (fig 3b), competitive in vitro IPs (fig 3c), SPR competition (fig 3d-g), and cryoEM (fig 4) that all independently validate this conclusion.

2. How do the authors interpret the lack of change in RU in e) when eIF2-P addition follows NSs and vice-versa in g)? Can NSs can only prevent inhibition of eIF2B by eIF2-P if it binds first as eIF2-P:eIF2B is very stable?

The lack of an increase in RU in both fig 3e and 3g when the second analyte is added is indicative of competition. This is a near-textbook example of competitive binding as assessed by SPR (Cytiva has a nice explanation of how these assays are set up: <https://www.cytivalifesciences.com/en/us/solutions/protein-research/knowledge-center/epitope-binning>). In this setup, we first saturate the

immobilized protein with either eIF2, eIF2-P, or NSs. Then we continue to flow this same concentration of material over the immobilized eIF2B as well as a second analyte. A further increase in signal indicates that the two proteins do not compete for the same binding site. A lack of an increase indicates that there is competition (since the binding site is saturated and the new analyte has nowhere to bind). This is not a reaction at equilibrium. But at equilibrium “[binding] first” isn’t really relevant. Both eIF2-P and NSs will come on and off in concordance with their on / off rates. Those values and the relevant concentrations of the eIF2-P and NSs during infection will determine how much eIF2-P vs NSs occupies the binding site. This experiment is sufficient to show that NSs can compete with eIF2-P for binding.

3. Does the more modest increase in RU in f) following NSs addition than seen for the reverse binding in d) mean that less NSs is bound to eIF2-eIF2B than to eIF2B alone here?

This is an overinterpretation of the data. Absolute RU levels should not be compared between these panels as different amounts (and different preps) of eIF2B are immobilized. Also, note that in 3f the second bump represents the NSs RU while in 3d the first bump represents the NSs RU. The relative ratios of captured NSs:eIF2 in the two traces is broadly comparable (~2x RU of eIF2 compared to NSs).

The descriptions in the results (lines 177-179) explaining these data should be expanded to make the findings clearer.

We have expanded this section to improve clarity.

4. The use of 'synergy' to describe the binding of NSs to eIF2B is not appropriate and adds nothing to the discussion, I suggest removing it. lines 330 and 503.

After reviewing the two instances of our use of 'synergistic' we have opted to keep the text as is. We feel that this properly conveys our point.

5. line 268 'independently' typo spotted.

Fixed

6. line 338 describes eIF2B as having tetrameric halves, when it has pentameric halves. Recent papers from the Walter lab and from the Ron lab show ISRIB does not 'staple' together decameric eIF2B, it is naturally decameric in vivo. ISRIB is primarily an allosteric activator. This sentence needs revising.

We have updated the sentence to include mention of eIF2B's A-State. We will note that although eIF2B is composed of 5 subunits, there is no evidence to support a formation of a pentamer. Instead, it appears that the complex can exist as subcomplexes (eIF2B $\beta\delta\gamma\epsilon$ tetramers or the constitutive eIF2B α_2 dimer) or the

fully assembled decamer which is composed of two tetrameric halves positioned on top of the eIF2B α_2 dimer. ISRIB binds across the dimerization interface and can either staple two tetramers into an octamer if eIF2B α_2 is limiting or staple the decamer in its active A-State.

Reviewer #3 (Remarks to the Author):

I acknowledge that the authors took into consideration most of my remarks from the first review.

I do not have further concerns at the present reviewing stage.